# Neuroligin-mediated neurodevelopmental defects are induced by mitochondrial dysfunction and prevented by lutein in *C. elegans*

Silvia Maglioni [1], Alfonso Schiavi [1,2], Marlen Melcher[3], Vanessa Brinkmann [1], Zhongrui Luo[4], Anna Laromaine [4], Nuno Raimundo [5], Joel N. Meyer[6], Felix Distelmaier[3] & Natascia Ventura [1,2✉]

Complex-I-deficiency represents the most frequent pathogenetic cause of human mitochondriopathies. Therapeutic options for these neurodevelopmental life-threating disorders do not exist, partly due to the scarcity of appropriate model systems to study them. *Caenorhabditis elegans* is a genetically tractable model organism widely used to investigate neuronal pathologies. Here, we generate *C. elegans* models for mitochondriopathies and show that depletion of complex I subunits recapitulates biochemical, cellular and neurodevelopmental aspects of the human diseases. We exploit two models, *nuo-5*/NDUFS1- and *lpd-5*/NDUFS4-depleted animals, for a suppressor screening that identifies lutein for its ability to rescue animals' neurodevelopmental deficits. We uncover overexpression of synaptic neuroligin as an evolutionarily conserved consequence of mitochondrial dysfunction, which we find to mediate an early cholinergic defect in *C. elegans*. We show lutein exerts its beneficial effects by restoring neuroligin expression independently from its antioxidant activity, thus pointing to a possible novel pathogenetic target for the human disease.

[1] IUF-Leibniz Research Institute for Environmental Medicine, 40225 Duesseldorf, Germany. [2] Institute for Clinical Chemistry and Laboratory Diagnostic, Medical Faculty, Heinrich Heine University, 40225 Duesseldorf, Germany. [3] Department of General Pediatrics, Neonatology and Pediatric Cardiology, University Children's Hospital, Heinrich Heine University, 40225 Duesseldorf, Germany. [4] Institut de Ciència de Materials de Barcelona, ICMAB-CSIC. Campus UAB, 08193 Bellaterra, Barcelona, Spain. [5] Department of Cellular and Molecular Physiology, Penn State College of Medicine, 500 University Drive, Hershey 17033, USA. [6] Nicholas School of the Environment, Duke University, Durham, NC 27708-0328, USA. ✉email: natascia.ventura@uni-duesseldorf.de

Over the last few decades, severe mitochondrial dysfunction, most often ascribed to genetic-mediated depletion of mitochondrial electron transport chain (ETC) complex subunits, has been established as the cause of numerous human diseases such as Leigh syndromes, Friedreich's ataxia or Parkinson's disease[1]. In these pathologies the most affected cells are typically those highly dependent on oxidative energy metabolism such as cardiac and skeletal muscle, pancreatic beta cells and, of course, neurons. Accordingly, mitochondriopathies often present with signs and symptoms that include myopathy, cardiac defects, neuronal abnormalities and sometimes diabetes or additional metabolic dysfunctions. Mitochondrial complex I (NADH:Ubiquinone-Oxidoreductase), the largest of the four mitochondrial ETC complexes, consists of 44 protein subunits encoded by the nuclear and mitochondrial genomes[2] and it constitutes the main entry point of electrons into the oxidative phosphorylation. Inherited genetic defects leading to human complex I (CI) deficiency are the most frequently encountered defects of the mitochondrial energy metabolism (with an incidence of about 1:8500 living births[3]) and typically cause early-onset neurometabolic disorders. NDUFS1 and NDUFS4 subunits are both part of the CI functional N-module, the structure responsible for the complex oxidoreductase activity. Numerous diseases-associated mutations in these two subunits, leading to reduced protein expression and/ or CI activity, have been described in human[4]. Leigh syndrome is a severe disorder ascribed to CI deficiency, which leads to progressive brain damage and muscular hypotonia[5]. Unfortunately, mitochondriopathies, due to their complexity and variety, are difficult not only to diagnose but also to treat, having very often a fatal outcome, and this is in part due to the scarcity of appropriate model systems to study them.

A considerable number of human disease genes have homologs in the genetically tractable model organism *Caenorhabditis elegans*, which has already yielded important insights into disease-relevant molecular mechanisms of many of them[6,7]. Moreover, *C. elegans*' small size, body transparency, short life cycle and genetic manipulability along with the absence of ethical requirements, make this organism a unique and attractive 3R-compliant model for in vivo high-throughput screening to search for new therapeutics or disclose modifiers of a particular phenotype and their underlying mode of action[8–10]. Notably, *C. elegans* is a powerful model organism for the study of neuronal development and degeneration since 1/3 of its cells are neurons and its nervous system is well characterized both at functional and structural level. Accordingly, the nematode system is successfully used to study neuronal aging and age-associated diseases such as Alzheimer's or Parkinson's diseases[11,12], and to screen for neuroprotective drugs[13,14]. Yet, only few studies have exploited *C. elegans* to untangle the pathogenesis and identify possible new treatments for neurodevelopmental pathologies associated with mitochondrial dysfunction and this is at least in part due to the fact that genetic depletion of mitochondrial genes is lethal[15–18]. Moreover, whereas not all defects of a complex brain observed in neuronal pathologies can be recapitulated in *C. elegans*, this simple (yet multicellular) model organism may be especially relevant to gain insight into early pathomechanistic consequences of mitochondrial dysfunction overseen in the human diseases, which in most cases present already with overwhelmed and irreversible neurometabolic deficits.

In this study we thus follow up on our previous findings revealing a clear threshold effect in determining the opposite—beneficial (e.g. lifespan extension) vs detrimental (e.g. larval arrest)—phenotypes observed upon different degrees of mitochondrial ETC subunits silencing (mild vs strong depletion respectively)[17,19,20], to develop *C. elegans* models for mitochondriopathies, especially linked to severe depletion of CI subunits.

We exploit the models to systematically characterize their early biochemical, cellular and neurobehavioral features. Of note, the discrete and very reproducible phenotypic features resulting from different degrees of mitochondrial dysfunction can be automatically quantified and are utilized in this study for a suppressor screening in search of new potential disease suppressors. We identify lutein, among a small library of natural compounds, for its ability to significantly suppress the arrest development and neuronal defects in *nuo-5*(*NDUFS1* homolog)- and *lpd-5*(*NDUFS4* homolog)-depleted animals.

Most notably, focusing on NDUFS1, for which there are no other animal models available, we pinpoint lutein protective mechanism through suppression of a neuroligin-dependent cholinergic synaptic defect, which we disclose in the *nuo-5*-depleted animals. We also confirm altered expression of synaptic neuroligins in the brain of a *Ndufs4* mutated mouse and find that lutein can reduce reactive oxygen species (ROS) production in patients' fibroblasts carrying *NDUFS1* or *NDUFS4* mutations. Our findings support the validity of exploiting *C. elegans* as a preclinical model to study pathomechanistic aspects of diseases associated with severe mitochondrial dysfunction, in particular with CI subunit depletion, and, most notably, point to novel potential targets and therapeutics for the treatment of the corresponding human pathologies.

## Results

**Development of *C. elegans* models for human mitochondriopathies**. To develop models for mitochondriopathies we initially carried out a cross-reference research through public-available databases (i.e. WormBase, OMIM, PubMed) to find as many as possible *C. elegans* annotated genes orthologous to nuclear-encoded genes which when mutated in humans lead to severe mitochondrial dysfunction and consequent neurodevelopmental pathologies (Table 1; Supplementary Table 1). We ended up with 41 genes for which we obtained sequence-verified dsRNA clones. Two independent rounds of RNAi screen were carried out for two consecutive generations in search of those clones giving the characteristic phenotypic effects associated with different degrees of mitochondrial dysfunction (Fig. 1a; Table 1)[19,20]: (i) a "mild" gene suppression causing slight decrease in size and fertility and slow development—phenotypes previously associated with the induction of beneficial pro-longevity mitochondrial stress responses; and (ii) a "strong" gene suppression inducing animals' sterility, growth arrest at the L2/L3 larval stage or lethality, reflecting the deleterious effect ascribed to severe mitochondrial dysfunction. Most of the clones screened showed a "mild" phenotype in the parental generation and a "strong" phenotype in the next generation (Fig. 1b, P0 vs F1 generations). When a "strong" phenotype was already observed in the parental generation, animals were fed with a diluted dsRNA-expressing bacterial clone to obtain a "mild" phenotype (Fig. 1c, diluted vs undiluted bacteria). 20 clones (Table 1) gave consistent and very reproducible results on animals' development and fertility upon different degrees of silencing and were selected for further characterization (while the remaining 21 clones were not further investigated, Supplementary Table 1).

Interestingly, when lifespan was analyzed, most of the genes, with a few exceptions (e.g. *pdr-1*), still elicited a pro-longevity effect (sometime even greater) when more severely suppressed, although with clear trade-off outcomes on fertility or development as actually expected by a disease state (Supplementary Fig. 1; Supplementary Table 2 and Supplementary Data 1). This implies that, consistent with our recent observation[21], health-related features such as fertility and development reflect the severity of animals' mitochondrial disruption better than lifespan

**Table 1 List of genes whose different degree of silencing consistently led to the typical *Mit* phenotypes.**

| Cosmid ID | Gene | Human ortholog, short gene description | Disease | RNAi Power | Phenotype |
|---|---|---|---|---|---|
| ZK973.10 | lpd-5 | NDUFS4, NADH-Dehydrogenase, Fe-S protein complex I subunit | Complex I deficiency (Leigh Syndrome) | Mild<br>Strong | Few progeny<br>Slow development; few eggs |
| Y45G12B.1 | nuo-5 | NDUFS1, NADH-Ubiquinone Oxidoreductase Fe-S Protein 1; complex I | Complex I deficiency (Leigh Syndrome) | Mild<br>Strong | Pale, small, few eggs<br>Slow development, arrested L2/L3 |
| C09H10.3 | nuo-1* | NDUFV1, mitochondrial NADH Dehydrogenase Ubiquinone Flavoprotein 1 | Complex I deficiency (Leigh Syndrome) | Mild<br>Strong | Pale, thin, sterile<br>Arrested as L2/L3 |
| T20H4.5 | T20H4.5* | NDUFS8, ubiquinone oxidoreductase core subunit S8 | Complex I deficiency (Leigh Syndrome) | Mild<br>Strong | Pale, small, few eggs<br>Arrested L2/L3 |
| F53F4.10 | F53F4.10 | NDUFV2, Fe-S complex I | Complex I deficiency Parkinson's Disease Susceptibility | Mild<br>Strong | Slow development<br>Slow development, arrested L2/L3 |
| C01F1.2 | sco-1 | SCO-1, SCO cytochrome c oxidase assembly protein 1 | Cytochrome-c oxidase deficiency disease | Mild<br>Strong | Pale, few progeny<br>Slow development, sick, thin, sterile |
| H14A12.2 | fum-1 | FH, fumarate hydratase, predicted to have fumarate hydratase activity | Fumarase deficiency, Leigh syndrome | Mild<br>Strong | Few eggs<br>Few progeny, slow development |
| T12E12.4 | drp-1 | DNM1L (dynamin 1 like) | Optic dystrophy | Mild<br>Strong | Similar to control<br>Small, pale |
| D2013.5 | eat-3* | OPA-1 or mgm-1, dynamin family GTPase | Dominant optic atrophy | Mild<br>Strong | Thin<br>Thin, slow development |
| F54H12.1 | aco-2 | ACO2, Mitochondrial aconitase | Infantile cerebellar-retinal degeneration and optic atrophy 9 | Mild<br>Strong | Few progeny<br>Sterile |
| Y47G6A.10 | spg-7* | AFG3L2, AFG3 like matrix AAA peptidase subunit 2 | Spinocerebellar ataxia (SCA28) | Mild<br>Strong | Pale, small, thin<br>Slow development, arrested L2 |
| F23B12.5 | dlat-1* | DLAT, dihydrolipoamide S-acetyltransferase | pyruvate decarboxylase deficiency | Mild<br>Strong | Sterile, slow moving<br>Sterile |
| K08E3.7 | pdr-1* | PARK-2 or parkin, E3 ubiquitin ligase | Parkinson's disease | Mild<br>Strong | Slow moving, pale, few progeny<br>Slow moving, pale, few eggs, protruding vulva |
| F25B4.6 | hmgs-1* | HMGCS1 & HMGCS2, 3-hydroxy-3-methylglutaryl-CoA synthase 1 & 2 | Mitochondrial HMG-CoA synthase deficiency | Mild<br>Strong | Pale, thin, slow moving<br>Sick, pale, slow moving |
| C15F1.7 | sod-1 | SOD-1, Cu-Zn SOD1 | Amyotrophic lateral sclerosis | Mild<br>Strong | Small<br>Similar to control |
| T22B11.5 | ogdh-1 | OGDH, predicted to have oxoglutarate dehydrogenase (succinyl-transferring) activity | Alpha ketoglutarate deficiency | Mild<br>Strong | Thin, pale, sterile adults<br>Thin, pale, slow development, sick |
| W02F12.5 | dlst-1* | DLST, dihydrolipoamide S-succinyl transferase | Possible cause of familial Alzheimer's disease | Mild<br>Strong | Skinny, pale<br>Long, pale |
| Y46G5A.2 | cox-10 | COX10, complex IV farnesyl transferase | Mitochondrial complex IV deficiency | Mild<br>Strong | Few eggs, pale<br>Pale, few eggs |
| T27E9.1 | tag-61* | SLC25A5, solute carrier family 25 member 5, mitochondrial adenine nucleotide transporters | Mitochondrial phosphate carrier deficiency | Mild<br>Strong | Thin, pale<br>Small, pale, sterile, protruding vulva |
| F101.12 | aldo-2 | ALDOB, encodes a fructose-bisphosphate aldolase | Hereditary fructose intolerance | Mild<br>Strong | Pale<br>Sick, pale |

Underlined genes are complex I subunits giving a typical *Mit* mutants phenotype in our RNAi screen, which we mainly followed up on in this study.
*These clones showed a strong phenotype already in the parental generation (P0, see Fig. 1) when used undiluted, and the mild effect was thus achieved by diluting the dsRNA expressing bacteria (P0, 1/10 or 1/15). Lifespan for these clones were carried out accordingly in the parental generation (P0) left undiluted or diluted (Supplementary Table 2).
For all the other clones a mild phenotype was observed in the undiluted parental generation (P0), while a strong one in the first filial generation (F1). Accordingly, lifespan with these clones were completed with P0 & F1 on undiluted dsRNA expressing bacteria (Supplementary Data 1). Nonetheless, since clones T20H4.5, *eat-3* and *spg-7* displayed a significant phenotype (but not as strong as others) already in the parental generation, their lifespan was assessed in P0/F1 as well as in diluted/undiluted conditions (see Supplementary Table 2 and Supplementary Data 1).

outcomes. Of note, these *C. elegans* detrimental phenotypes are typically associated with severe mitochondrial damage since in this stage mitochondria are necessary in the gonad to start germline expansion and in the nervous system to progress through development[22,23]. Given their very strong and reproducible phenotype (L2/L3 larval arrest) and health importance in humans (disease prevalence)[4], we then decided to focus on animals with severe suppression of CI subunits (underlined genes

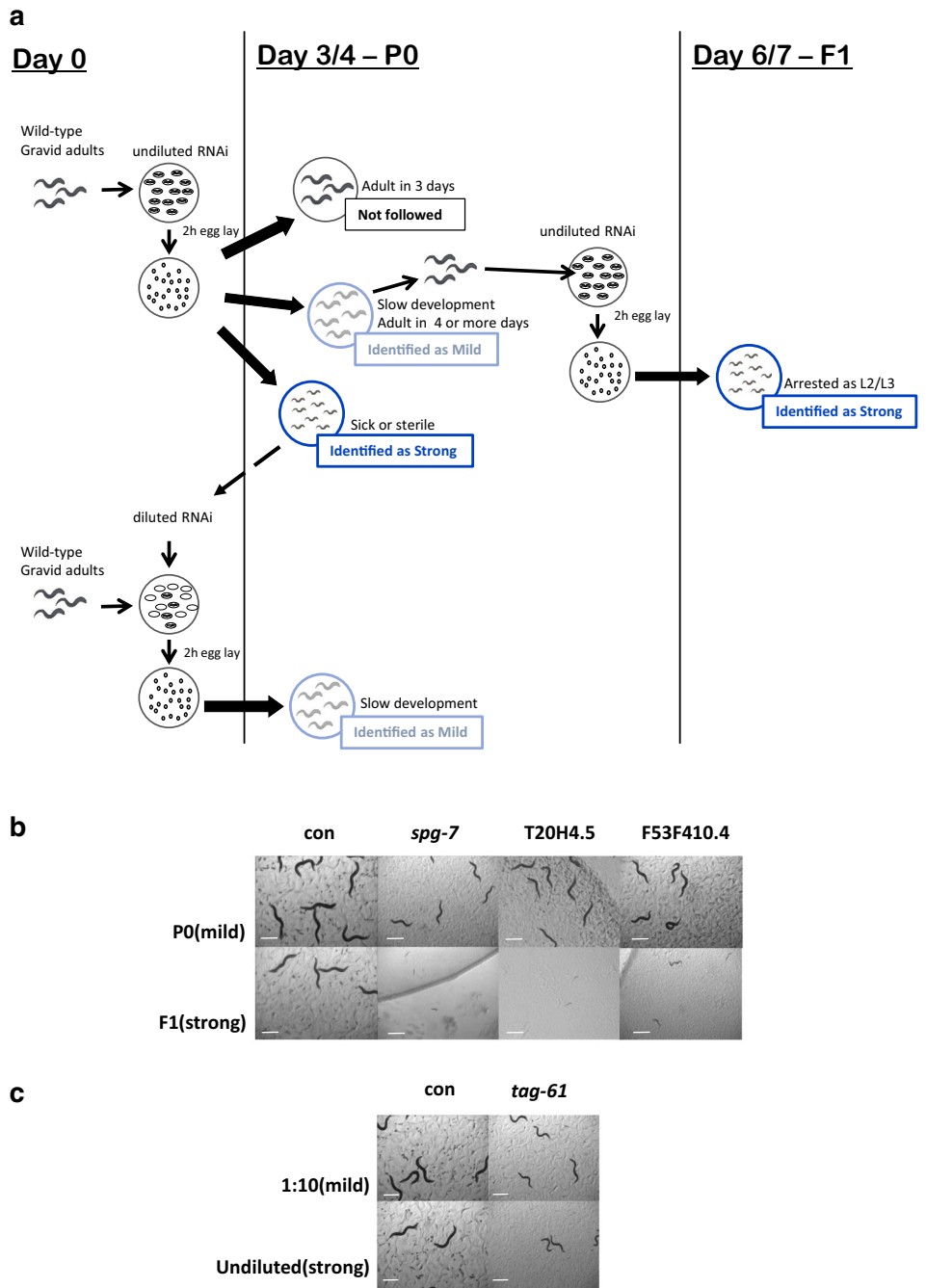

**Fig. 1 Phenotypic screening to identify *C. elegans* models for human mitochondrial-associated disorders. a** At Day 0 animals were allowed to lay eggs on undiluted bacteria expressing dsRNA for each clone under study. Progeny development was observed for the following 3/4 days (P0 or parental generation). Three possible scenarios were then identified for the different RNAi clones. (i) No effect (black): worms reached adulthood in 3 days and looked similar to wild-type, control (empty-vector) treated worms. These clones were not subsequently followed. (ii) A "mild" phenotype (light blue): decreased size and fertility and slow development were observed (representative images are shown in panels **b**, **c**). (iii) A "strong" phenotype (dark blue): sterility, growth arrest or lethality were observed (representative images are shown in panels **b**, **c**). The clones giving a "mild" phenotype were followed in the next generation, in search of a strong effect (Day 6/7, F1 or first filial generation). If a strong phenotype was instead observed in the P0 then the RNAi bacteria were diluted with bacteria expressing empty vector (1:10 or 1:50), an egg lay was performed and the progeny was followed again in the P0 to identify a mild effect. Screening representative pictures of different phenotypes induced by mild suppression of the listed genes (P0 in panel **b** or 1:10 dilution in panel **c**) or strong (F1 in panel **b** or undiluted in panel **c**), after 4 days from egg-lay. Mild genes suppression reduces adult animals' size and fertility rate. Strong gene suppression leads to animals early developmental arrest (F1, **b**) or sterility (undiluted, **c**). Two of the clones shown in **b** (i.e. *spg-7* and T20H4.5), already led to a significant effect in the parental generation (P0), and were therefore also diluted to achieve an even milder phenotype more similar to those shown in panel **c**. See also Supplementary Tables 1, 2 and Supplementary Data 1. See text for details. Pictures were acquired with a Leica MZ10 F modular stereo microscope connected to a color digital camera. Scale Bars 500 μm. Five biologically independent experiment with similar results.

in Table 1) and further characterized features relevant for the human disorder.

**Severe suppression of Complex I subunits impairs neuronal and mitochondrial functionality.** Most CI defects are associated with Leigh Syndrome, a devastating neuromuscular degenerative disorder with early childhood onset and very little, and only symptomatic, therapeutic options[5]. Indeed, due to the complexity of the human brain, in most cases the disease already presents with overwhelming neurometabolic abnormalities, which do not allow effective treatment and quickly lead to an inevitable fatal outcome. We thus exploited the power of nematode behaviors to reveal possible early neuromuscular functional defects induced by mitochondrial deficiency. Using different chemicals, which in *C. elegans* are sensed by different sub-classes of neurons, we assessed animals chemotaxis, a very sensitive readout of animal health[24], and found that strong suppression of the different CI subunits caused a significant neuronal impairment compared to stage-matched L3 larvae controls (Fig. 2a, b; Supplementary Fig. 2a–d; Supplementary Table 2). Besides the neurodevelopmental defects CI deficiency typically manifests with rapid deterioration of the motor function. Interestingly, severe suppression of different CI subunits did not significantly reduce animals' ability to move in solid agar plate compared to wild-type animals (Supplementary Fig. 2e), with the exception of one clone, *nuo-1*. Yet, notably, similar to the perceived fatigue often observed in patients with mitochondrial disorders[25], a locomotion defect could be revealed under energetic challenge, a condition that in *C. elegans* can be simulated by swimming[26] (Supplementary Fig. 2f).

For further characterization of the models, we then narrowed our study to two CI subunits, which are both part of the CI functional N-module, the structure responsible for oxidation of NADH to NAD$^+$ (the main CI activity), and specifically NUO-5 (NDUFS1) and LPD-5 (NDUFS4). While their suppression gave similar phenotypes the former represent one of the N-module so-called core subunits, which are conserved from bacteria to human, while the latter is one of the accessory subunits, which have been shown to vary between species[27]. NDUFS1 is a 75-kDa subunit essential for the enzymatic function, specifically for the electron transfer within CI and is thought to be the first of many Fe-S proteins to accept electrons from an NADH-flavoprotein reductase[28]; LPD-5 is instead an 18-kDa subunit, which is crucial for the assembly and stability of the complex and it is predicted to have oxidoreductase activity[29]. Symptoms associated with CI deficiency can be concurrently ascribed to cell-autonomous and non-autonomous effects of the mitochondrial dysfunction. Animals L3 arrest implies the RNAi is likely working in the nervous system – as in this stage mitochondria are necessary in the nervous system to progress through development[23]—but *C. elegans* gene silencing can be less efficient in neurons. Thus, to address the contribution of neuronal vs non-neuronal cells to animals' pathology, we compared the effect of *nuo-5* and *lpd-5* silencing on development in the wild-type strain (WT) with that in a neuronal-specific sensitive strain (TU3401) and in a strain with overall increased sensitivity to RNAi (but especially in neurons) (CL6114). WT and TU3401 animals develop normally and in control conditions all embryos reach the fertile stage in 72 h. Instead, CL6114 animals have a slower developmental rate and all animals in control conditions become gravid adults after 96 h. Interestingly, the percentage of animals reaching the fertile stage upon silencing was similar in N2 and TU3401, suggesting that the observed developmental defects are primarily ascribed to neuronal gene suppression. Instead, the effect of the RNAi appeared to be stronger in the CL6114 since animals never become fertile adults (Supplementary Fig. 2g), suggesting that

systemic mitochondrial suppression has broader repercussion than neuronal silencing. Thus, to closely recapitulate the systemic mitochondrial disease state and avoid possible confounds due to strains genetic background and developmental timing, we decided to continue our characterization using systemic RNAi in the wild-type *C. elegans* strain.

Since cardiopulmonary failure is most often the underlying cause of mortality in patients with mitochondrial disorders, we characterized animals pharyngeal muscle functionality. Indeed, similarities between *C. elegans* pharyngeal muscle and the vertebrate heart in terms of anatomic development and physiology have been described, which suggest convergent evolution between two autonomous muscular pumps with similar biological roles[30]. Strikingly, automatic measurement through a microfluidic device[31] revealed that *nuo-5* and *lpd-5* RNAi-treated animals have an abnormal electropharyngeogram (EPG) with significantly decreased pump frequency (Fig. 2c) and increased average inter-pump interval (IPI) as well as R to E ratio when compared to control nematodes (Supplementary Fig. S2h, i).

Mitochondrial respiration was then also measured with the Seahorse Bioscience Flux Analyzer, optimized for nematodes at L2/L3 larval stage[21]. Consistent with a mitochondrial disease state, most parameters associated with mitochondrial respiration were significantly affected upon severe suppression of *nuo-5* or *lpd-5* compared to wild-type animals. Specifically, basal oxygen consumption rate (OCR), maximal respiratory capacity (upon treatment with the uncoupling agent carbonyl cyanide 4-trifluoromethoxy-phenylhydrazone), ATP-linked OCR (in the presence of the ATP inhibitor Dicyclohexylcarbodiimide) as well as steady-state ATP levels, were halved compared to control animals (Fig. 2d–g). Spare respiratory capacity (resulting by subtracting basal from maximal OCR), which indicates organism's ability to respond to increased energy demand, and proton leak, were however only significantly reduced in *nuo-5*-depleted animals (Supplementary Fig. 3a, b). Interestingly, *nuo-5* and *lpd-5* RNAi significantly increased ROS production as expected (Fig. 2h), but did not induce any major sign of oxidative damage, at least at this early stage: the expression of a classical oxidative stress sensor (p38MAPK/PMK-1) was not affected (Supplementary Fig. 3c) and actually both lipid and amide oxidation were diminished (Supplementary Fig. 3d, e). Moreover, although mitochondrial DNA is especially sensitive to oxidative damage[32], *nuo-5* and *lpd-5* RNAi-depleted animals only presented with nuclear but not mitochondrial DNA damage (Supplementary Fig. 3f, g) and, quantitative measurements of mitochondrial genome copy number, which can be oxidative stress-responsive[33], also did not reveal any significant difference between control and RNAi-treated animals (Supplementary Fig. 3h). These results suggest the induction of stress response defensive mechanisms. Accordingly, *nuo-5* and *lpd-5* RNAi increased the expression of genes involved in mitochondrial quality control pathways[17,34]: antioxidant response (glutathione-S-transferase, *gst-4*), mitochondrial unfolded protein response (heat-shock protein, *hsp-6*), and mitophagy (BCL2/adenovirus E1B 19-kDa-interacting protein 3, *BNIP3* homolog, *dct-1*) (Fig. 2i–k; Supplementary Fig. 4a). Disturbing *C. elegans* mitochondrial function also induces autophagy, lipid remodeling[35] as well as drug detoxification and pathogen-response genes[36]. The nuclear translocation of HLH-30, a common regulator of these 'cytoprotective' pathways[37], was also significantly increased in *nuo-5* and *lpd-5* arrested L3 animals (Fig. 2l; Supplementary Fig. 4b), differently from pro-longevity *daf-2* RNAi that only promoted HLH-30 activation in adults but not in the L3 larvae (Supplementary Fig. 4c). This suggests that differently from the beneficial activation of TFEB/HLH-30 induced by DAF-2 suppression[37], activation of this transcription factor early in life is rather associated with detrimental effects.

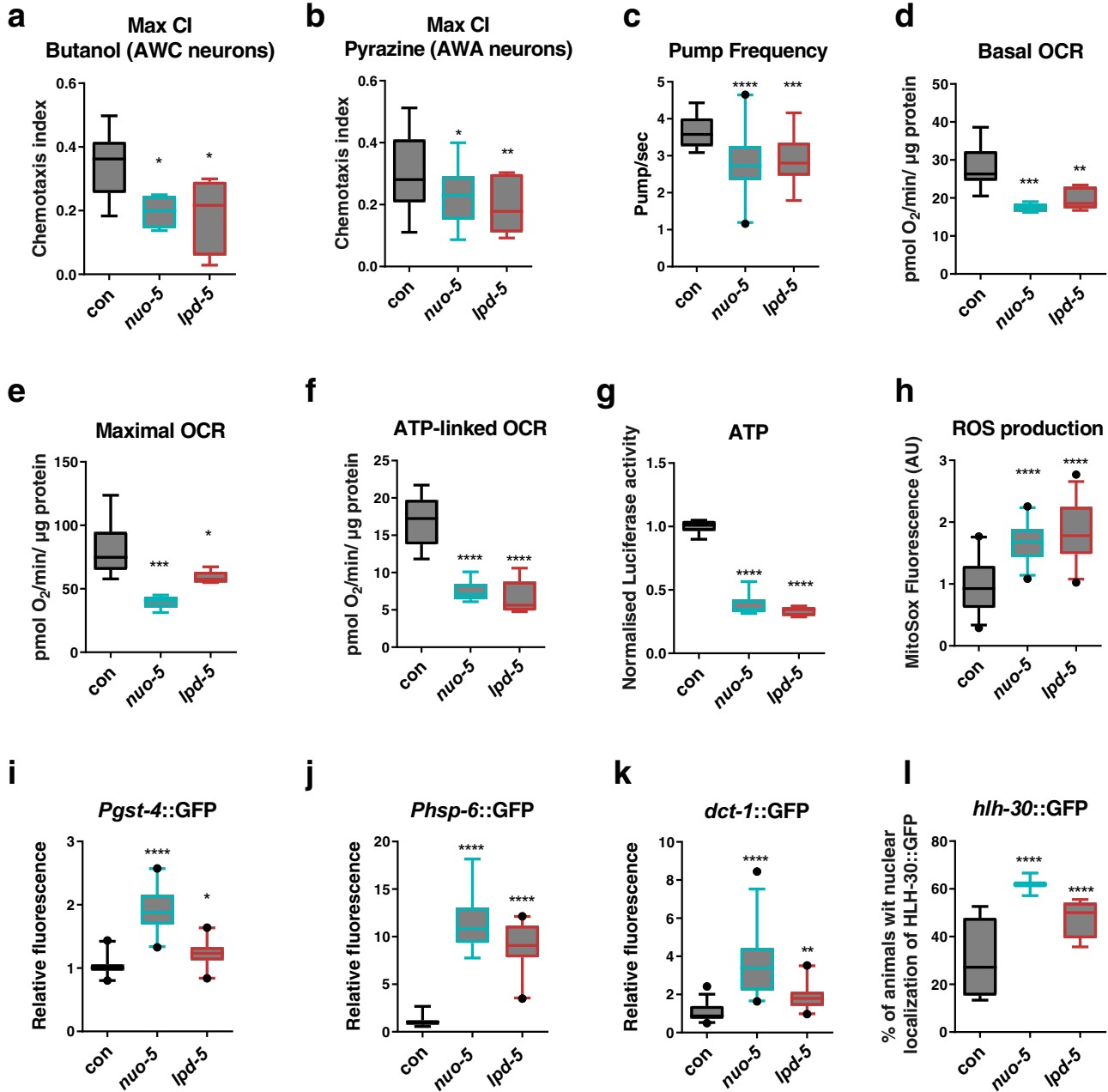

**Fig. 2 Severe suppression of Complex I genes significantly impairs sensory neurons and mitochondrial functionality.** Animals were fed bacteria transformed either with empty-vector (con) or with vector expressing dsRNA against *nuo-5* or *lpd-5*. **a, b** Chemotaxis index reached after 240 minutes in response to 1% Butanol ($n \geq 630$) or 1% Pyrazine ($n \geq 610$), over three biologically independent experiments. *p* values in **a**: con vs *nuo-5* p = 0.024, con vs *lpd-5* p = 0.019. *p* values in **b**: con vs *nuo-5* p = 0.032, con vs *lpd-5* p = 0.0082. **c** Pharyngeal Pump frequency was measured using the ScreenChipTM System (InVivo Biosystems). $n \geq 30$ over two (for *lpd-5*) or four (for *nuo-5*) biologically independent experiments. con vs *nuo-5* p < 0.0001, con vs *lpd-5* p = 0.0003. **d–f** Basal OCR, maximal OCR and ATP-linked OCR were measured using the Seahorse XF24 Analyzer. n $\geq$ 3750 over three biologically independent experiments. *p* values in **d**: con vs *nuo-5* p = 0.0007, con vs *lpd-5* p = 0.0039. *p* values in **e**: con vs *nuo-5* p = 0.0003, con vs *lpd-5* p = 0.0319. *p* values in **f**: con vs *nuo-5* p < 0.0001, con vs *lpd-5* p < 0.0001. **g** ATP production was measured using the luciferase-expressing strain PE255 $n \geq 1000$ over three biologically independent experiments. ****p* values < 0.0001. **h** ROS production was measured with MitoSOX Red, $n \geq 50$ over two biologically independent experiments. ****p* values < 0.0001. **i** P$_{gst-4}$::GFP expression, $n \geq 65$ over four biologically independent experiments. con vs *nuo-5* p < 0.0001, con vs *lpd-5* p = 0.022. **j** P$_{hsp-6}$::GFP expression, $n \geq 50$ over four biologically independent experiments. ****p* values < 0.0001. **k** P$_{dct-1}$::DCT-1::GFP expression, $n \geq 36$ over two biologically independent experiments. *p* values: con vs *nuo-5* p < 0.0001, con vs *lpd-5* p = 0.0072. **l** P$_{hlh-30}$::HLH-30::GFP expression, n $\geq$ 56 over four biologically independent experiments. ****p* values < 0.0001. In all figure's panels box plots indicate median (middle line), 25th, 75th percentile (box) and 5th and 95th percentile (whiskers) as well as outliers (single points). One-way ANOVA. Data for all panels are provided as a Source Data file.

Taken together, characterization of our models indicates that, similar to the human diseases, animals with CI subunits depletion display severe mitochondrial, cellular, developmental and neuromuscular defects.

**A phenotype-based screening identifies Lutein as a Complex I disease suppressor.** In vivo genetic and compounds screening with model organisms exponentially grew in the past decade as strategies to identify modifiers of specific phenotypes ranging from simple proteins expression to more complex animals' behaviors such as aging or disease-associated features[6,8,38]. Nutraceuticals supplemented through the diet may provide more feasible therapeutic opportunities than genetic or pharmacological interventions. We thus specifically took advantage of the reproducible, discrete and automatically quantifiable disease-relevant phenotype – developmental arrest – observed upon severe mitochondrial dysfunction, to screen a small library of mainly natural compounds (available in the lab and with previously described anti -aging, -bacterial, -inflammatory, -oxidant, and -cancer activity; Supplementary Data 2) in search of potential disease suppressors. The developmental arrest induced by severe knockdown of CI proteins might however be irreversible or require a treatment at a specific stage in order to be prevented or rescued. Thus, we first used two different approaches to have a proof of principle of the feasibility of our suppressor screen. In the first approach we took advantage of the p53 C. elegans mutant, cep-1(lg12501), which can prevent the developmental arrest induced by severe suppression of nuo-2 and of other mitochondrial proteins[20]. Lack of cep-1 also partially rescued the development arrest observed in different disease models (Supplementary Table 4), indicating the arrest is reversible. In the second approach, we moved the F1 progeny from RNAi-expressing- to empty vector-expressing- bacteria at different stages, namely from eggs to L3. Depending on the disease model, the developmental arrest could be prevented moving the animals on control bacteria from eggs to L2, but in most cases not after L3 arrest (Supplementary Fig. 5a; Supplementary Table 4), with obvious implications for the efficacy of a possible therapeutic approach.

A total of 37 compounds were then supplemented at different concentrations, based on previous studies showing beneficial effects in different systems (and ranging from 0.5 μM to 1 mM), to the second generation of animals (starting from eggs) fed UVB-killed bacteria transformed with either control vector or with nuo-5 or lpd-5 dsRNA (Supplementary Fig. 5b). Among tested conditions (compounds & doses), 18 met our arbitrary cutoff, 11 in nuo-5 RNAi and 7 in lpd-5 RNAi treated animals, leading to a total of 12 compounds which allowed more than 20% of worms to develop into fertile adults after 6 days of treatment (Fig. 3a–c; Supplementary Fig. 6a, b; Supplementary Table 3), compared to wild-type animals, which developed into adults in 3 days. Significant differences were achieved only with isovitexin 10μM on nuo-5, with kahalalide F 0,5μM on lpd-5 and with lutein 1μM on both. Moreover, out of twelve compounds only three, i.e. lutein, isovitaxin and macrosporin, were effective against both RNAi (Fig. 3a–c). We then looked at the effect of these three compounds on development from day 4 to 6 and observed that the treatment with lutein was the only one reproducibly showing significant effects across different time points (Supplementary Fig. 7a–c).

To avoid misleading interpretation of compounds effects we then excluded that most relevant compounds did not generically (i.e. skn-1 or phi-7 RNAi-induced embryonic lethality, Supplementary Fig. 7d, e) or specifically (i.e. nuo-5 transcript expression, Fig. 3d) affect the efficacy of the RNAi. Finally, since the energetic

defect as well as the reduced pharyngeal pumping activity upon nuo-5 and lpd-5 depletion may impact on compounds absorption and/or metabolism we measured lutein accumulation inside the animals. Importantly, through infrared micro-spectroscopy we could detect lutein absorbance peak at 1515.8 cm$^{-1}$[39] in the worms (Fig. 3e), confirming that animals uptake the drug independently of their mitochondrial deficiency and also through ways other than just oral uptake e.g. through the cuticle or via sensory neurons exposed to the environment. We thus identified lutein as a nutraceutical able to significantly and specifically prevent the developmental arrest induced by systemic mitochondrial deficiency.

Lutein is a carotenoid belonging to the xanthophyll family that is found abundantly in leafy dark green vegetables, egg yolk, animal fat and human eye retinal macula[40]. Increasing evidence points towards the beneficial effects of lutein in several pathological conditions, ranging from cancer to neurodegenerative disorders[41–43]. Strikingly, we found that lutein not only partially rescued the developmental defect of nuo-5 or lpd-5 depleted animals (Fig. 3c), but also significantly ameliorated their neuronal deficits (Fig. 3f–i). Specifically, lutein fully suppressed the chemotaxis defect induced by nuo-5 and lpd-5 RNAi towards different chemicals, in some cases even above the level of wild-type lutein-treated animals. Indeed, lutein enhanced the sensory neuron functionality in the wild-type animals, but this improvement was greatly accentuated in the nuo-5 and lpd-5 depleted animals (Fig. 3f–i). Importantly, lutein also suppressed the pharyngeal pumping defect observed in the disease models (Fig. 3j; Supplementary Fig. 7f).

The most likely physiological functions of this pigment are protection against oxidative- and radiation- induced damage of membrane's lipids[44]. Accordingly, lutein was able to decrease ROS in nuo-5- and lpd-5 depleted worms (Fig. 3k) and completely prevented HLH-30 activation (Fig. 3l), which may indeed be increased either by ROS or mitochondrial damage. Of note, lutein also decreased ROS levels in fibroblasts derived from patients carrying mutations in the corresponding human homologs NDUFS1/nuo-5 and NDUFS4/lpd-5 (Fig. 3m; Supplementary Fig. 8a, b). However, while nuo-5 and lpd-5 depleted animals were more sensitive to the mitochondria ETC inhibitor sodium azide, lutein failed to protect against it (Supplementary Fig. 8c), it did not consistently suppress the induction of mitochondrial stress response genes (Supplementary Fig. 8d–g) and it did not modulate oxidation of cellular components (Supplementary Fig. 8h-i). Moreover, we tested the effect of other antioxidants with known effect on another C. elegans model for Complex I disease[16], namely N-Acetyl-cysteine (NAC), Vitamin C (VitC) and Coenzyme Q10(CoQ10) and found that while general antioxidants such as NAC and VitC, similar to lutein, prevented the developmental arrest of nuo-5 depleted animals, this was not the case with the mitochondrial antioxidant CoQ10 (Supplementary Fig. 8j). Taken together, results described so far revealed that lutein substantially ameliorates developmental, neuronal and cellular dysfunction upon depletion of CI subunits but most likely impinging on downstream mitochondrial-modulated processes rather than through direct beneficial effects on mitochondria.

**Lutein rescues a newly identified synaptic defect in NDUFS1/nuo-5-depleted animals.** To gain insight into the molecular mechanisms accounting for the deleterious outcomes induced by severe mitochondrial dysfunction and the protective effects of lutein, we (i) first compared the transcriptomic profile of stage-matched (L3 larvae) cohorts of wild-type animals with that of larvae subjected to mild or strong nuo-5 RNAi (Fig. 4a); (ii) and then compared the transcriptomic profile of strong nuo-5 RNAi

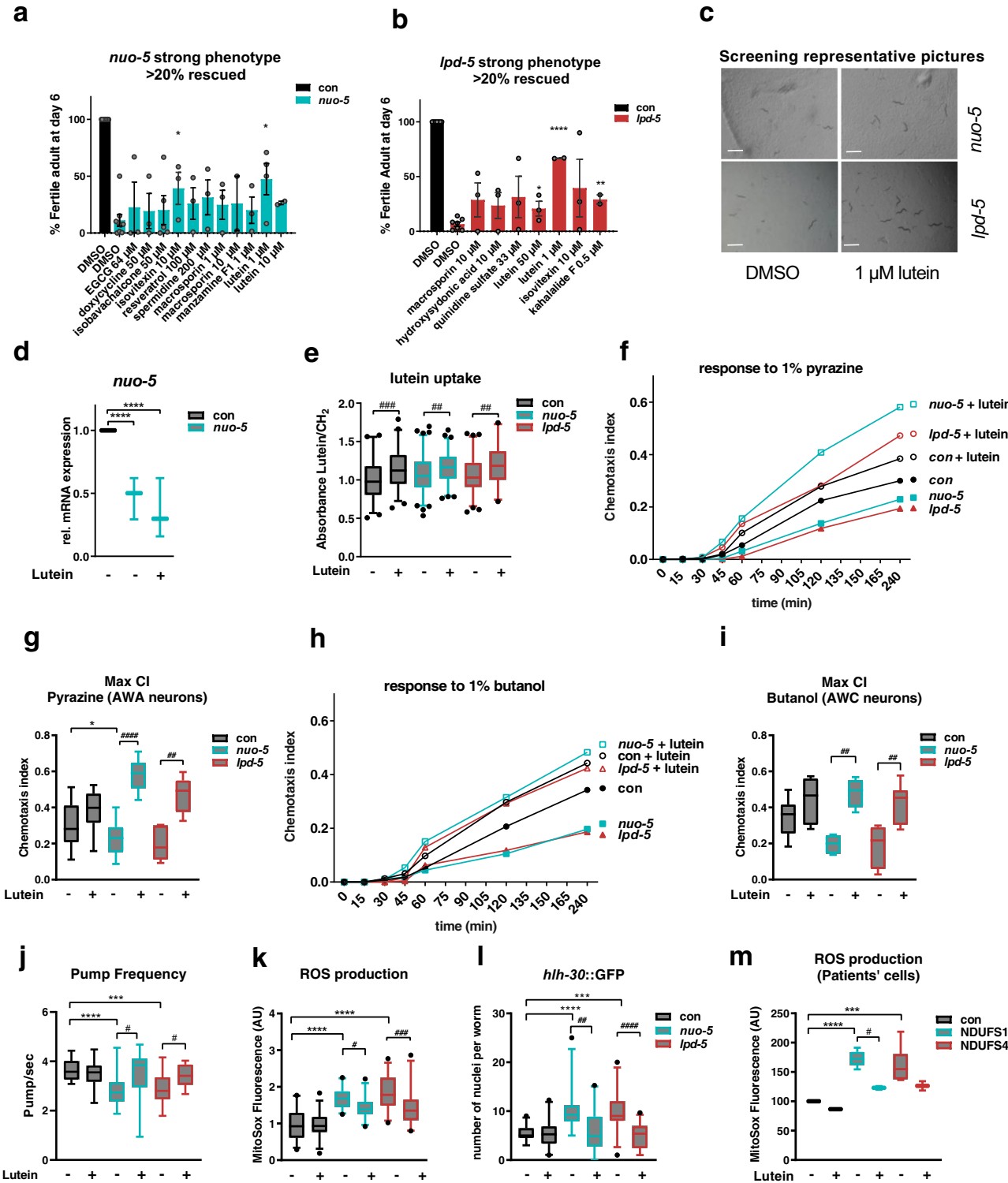

animals, treated and untreated with lutein, with that of animals fed empty vector control. Remarkably, mild mitochondrial stress led to very few changes in gene expression in the L3, in stark contrast with differences observed in the adults[45,46]. Only four genes, *fmo-2*, B0205.13, T16G1.6 and *nuo-5*, were indeed differentially expressed between control and mild *nuo-5* RNAi, based on a relatively liberal FDR-corrected $p < 0.05$ cut-off, with *nuo-5* expression being more severely suppressed upon strong RNAi. Moreover, consistent with the different degree of mitochondrial disruption[21], the 303 genes differentially expressed uniquely

between mild and severe *nuo-5* treatments all belong to GO terms associated with mitochondria (Fig. 4b, d, f; Supplementary Data 3), clearly indicating that the metabolic alterations we previously observed in response to mild mitochondrial stress[21] precede genomic reprogramming. Thus at least at this early developmental stage, the mitochondrial remodeling upon mild mitochondrial stress appears to successfully compensate for the genetic deficiency, obviating the need for detectable transcriptional changes. In contrast, large transcriptomic changes are triggered in larvae upon severe mitochondrial stress. Interestingly,

**Fig. 3 Phenotype-based screening identifies lutein ability to rescue behavioral and cellular defects in models of Human Complex I deficiency.**
**a–b** Compounds rescuing more than 20% of the population. Bar graphs represent means ± SEM and corresponding data points. $n \geq 40$ over two to five biologically independent experiments, two-tailed Student's $t$ test. $p$ values in **a**: nuo-5 DMSO vs isovitexin 10 μM $p = 0.0379$, nuo-5 DMSO vs lutein 1 μM $p = 0.0102$. $p$ values in **b**: lpd-5 DMSO vs lutein 50 μM $p = 0.022$, lpd-5 DMSO vs lutein 1 μM $p < 0.0001$, lpd-5 DMSO vs kahalalide F 0.5 μM $p = 0.001$.
**c** Representative pictures of 4 wells of a 12-well plate 3 days after eggs hatching. Scale bars, 1 mm, $n \geq 40$ over four biologically independent experiments.
**d** Relative mRNA expression of nuo-5 assessed by qPCR. One-way ANOVA. ****$p$ values < 0.0001. Three biologically independent experiments.
**e** Quantification of lutein ratio addressed with the use of peak 1515.775 cm$^{-1}$ as Lutein's characteristic peak and ratio equation (Lutein/CH$_2$). Normalized on control values, $n \geq 73$ over three biologically independent experiments. Box plots indicate median (middle line), 25th, 75th percentile (box) and 2.5th and 97.5th percentile (whiskers) as well as outliers (single points). p-values: con vs con + lutein $p = 0.0001$, nuo-5 vs nuo-5 lutein $p = 0,007$, lpd-5 vs lpd-5 + lutein $p = 0.005$. **f, h** Chemotaxis curves and **g, i** maximum chemotaxis index, reached after 240 minutes using as attractants **f, g** 1 μl of 1% pyrazine or **h, i** 1 μl of 1% butanol, $n \geq =100$ over three biologically independent experiments. $p$ values in **g**: con vs nuo-5 $p = 0.0412$, nuo-5 vs nuo-5 + lutein $p < 0.0001$, lpd-5 vs lpd-5 + lutein $p = 0.0015$. $p$ values in **i**: nuo-5 vs nuo-5 + lutein $p = 0.007$, lpd-5 vs lpd-5 + lutein $p = 0.006$. **j** Pharyngeal pump frequency in adult worms fed for one generation with RNAi, $n \geq 26$ over three biologically independent experiments, the two conditions were tested separately thus, unpaired one-way ANOVA was used. con vs nuo-5 $p < 0.0001$, con vs lpd-5 $p = 0.0003$, nuo-5 vs nuo-5 + lutein $p = 0.0469$, lpd-5 vs lpd-5 + lutein $p = 0.00182$.
**k** ROS production, measured with MitoSOX Red, quantified in the pharyngeal bulb, $n \geq 48$ over two biologically independent experiments. con vs nuo-5 $p < 0.0001$, con vs lpd-5 $p < 0.0001$, nuo-5 vs nuo-5 + lutein $p = 0.033$, lpd-5 vs lpd-5 + lutein $p = 0.0003$. **l** Quantification of GFP nuclear translocation in a strain expressing the P$_{hlh-30}$::HLH-30::GFP transgene, n $\geq 48$ over three biologically independent experiments. con vs nuo-5 $p < 0.0001$, con vs lpd-5 $p = 0.0003$, nuo-5 vs nuo-5 + lutein $p = 0.0049$, lpd-5 vs lpd-5 + lutein $p < 0.0001$. **m** ROS production in patients' derived skin fibroblasts after 24 h incubation with lutein. $n \geq 107$ over three biologically independent experiments. con vs NDUFS1 $p < 0.0001$, con vs NDUFS4 $p = 0.0002$, NDUFS1 vs NDUFS1 + lutein $p = 0.0357$. In **f–l** animals were fed bacteria transformed either with empty-vector (con) or with vector expressing dsRNA against nuo-5 or lpd-5 treated with lutein 1 μM (unless otherwise specified) or left untreated. Box plots indicate median (middle line), 25th, 75th percentile (box) and 5th and 95th percentile (whiskers) as well as outliers (single points), except otherwise indicated. Bar graphs represent means ± SEM. Asterisks (*) denote significant differences vs control, # denote differences among conditions. Two-way ANOVA followed by Tukey's multiple-comparisons test, unless otherwise specified. Data for all panels are provided as a Source Data file.

the 1152 genes uniquely different between control and strong nuo-5 suppression, revealed GO terms affecting multiple cellular compartments (Fig. 4c, e, g; Supplementary Data 3), from nucleus to cytoplasm to the Golgi apparatus and membrane components, as well as genes involved in lifespan determination, ribonucleoprotein complex biogenesis and neuropeptide signaling, thus displaying a more general cellular remodeling not only limited to mitochondria.

Accordingly, in the second transcriptomic analysis, the comparison between wild-type and strong nuo-5 depleted animals revealed 2274 altered genes (Fig. 5a), implicated in fundamental cellular morphogenesis processes and especially, as expected, in organismal and nervous system development (Supplementary Fig. 9a; Supplementary Data 4), reflecting the very strong effect of the treatment on worm's biology ultimately leading to growth arrest. On the other hand, interestingly, when comparing wild-type animals with nuo-5 treated with lutein (rescued animals), the number of altered genes diminished to 1296, indicating that roughly half of the genes were restored to wild-type levels. Notably, consistent with the rescue of the neurodevelopmental deficits observed upon lutein treatment, most of these genes belong to developmental and neuronal processes (Supplementary Fig. 9b; Supplementary Data 4). We then focused our attention on the differences between nuo-5 depleted worms left untreated or treated with lutein, and identified 143 differentially expressed genes (DEG, Fig. 5a, b; Supplementary Data 5). Analysis of nuo-5 expression in the microarray data supported previous qPCR results and confirmed that lutein-rescuing effects are specific and not ascribed to a reduced potency of the nuo-5 RNAi (Fig. 5c). Moreover, the heatmap (Fig. 5b; Supplementary Data 5) and the analysis of a sample of genes (Fig. 5d–f) demonstrated the high reproducibility of the data. Intriguingly, GO analysis (Supplementary Data 4) revealed four main molecular pathways enriched upon lutein treatment in nuo-5 depleted animals: (i) secretion (synaptic and neurotransmission regulation), (ii) endosomal transport, (iii) protein import into the nucleus and (iv) mitotic sister chromatid segregation (cell cycle) (Fig. 5g). In support of the cellular defects, no major signature relative to oxidative stress responses or damage were revealed in this early developmental

stage. Instead, we speculated that defective mitochondria-induced energetic failure negatively impacts on vesicle trafficking and consequently synaptic functionality, which is somehow rescued by lutein.

To address our hypothesis, we took advantage of behavioral assays commonly used to assess synaptic functionality in C. elegans[47–51]. Synaptic transmission mutants display altered resistance to aldicarb, an acetyl-cholinesterase inhibitor[48]: mutants with increased cholinergic transmission are hypersensitive to aldicarb and paralyze faster than wild-type animals (Hic phenotype), while mutants with reduced cholinergic transmission have increased resistance and take longer time to paralyze upon aldicarb treatment (Ric phenotype). Strikingly, in support of our hypothesis, nuo-5 depleted larvae paralyzed faster than their stage-matched wild-type in the presence of the same concentration of aldicarb, indicating an increased cholinergic transmission. Most notably, nuo-5 worms treated with lutein from eggs to L3 larvae lose the Hic phenotype and have a normal response to Aldicarb (Fig. 6a). Remarkably, we observed the same Hic phenotype upon silencing of other three disease-associated CI subunits, namely lpd-5, T20H4.5 and F53F4.1 (Table 1), and lutein significantly rescued the synaptic defect (Supplementary Fig. 10a–c). The Hic phenotype could be ascribed either to presynaptic alterations, such as inability to negatively regulate presynaptic ACh release or reuptake, or to post-synaptic defects, like loss of post-synaptic ACh regulation or inhibitory GABA transmission. Since the aldicarb assay does not distinguish between these different scenarios we took advantage of levamisole, a selective agonist of the nicotinic acetylcholine receptors found at the postsynaptic side of the neuromuscular junction (NMJ)[49]. Wild-type nematodes exposed to levamisole paralyze over time due to excessive excitation of the acetylcholine receptors and worms with loss-of-function mutations in these receptors are usually more resistant to levamisole. Instead, animals with mutations affecting presynaptic function paralyze either to a similar speed or slightly faster than wild-type on levamisole plates (due to a possible compensatory postsynaptic increase in cholinergic receptors expression or function)[52,53]. We observed that nuo-5 deficient worms also paralyzed faster than

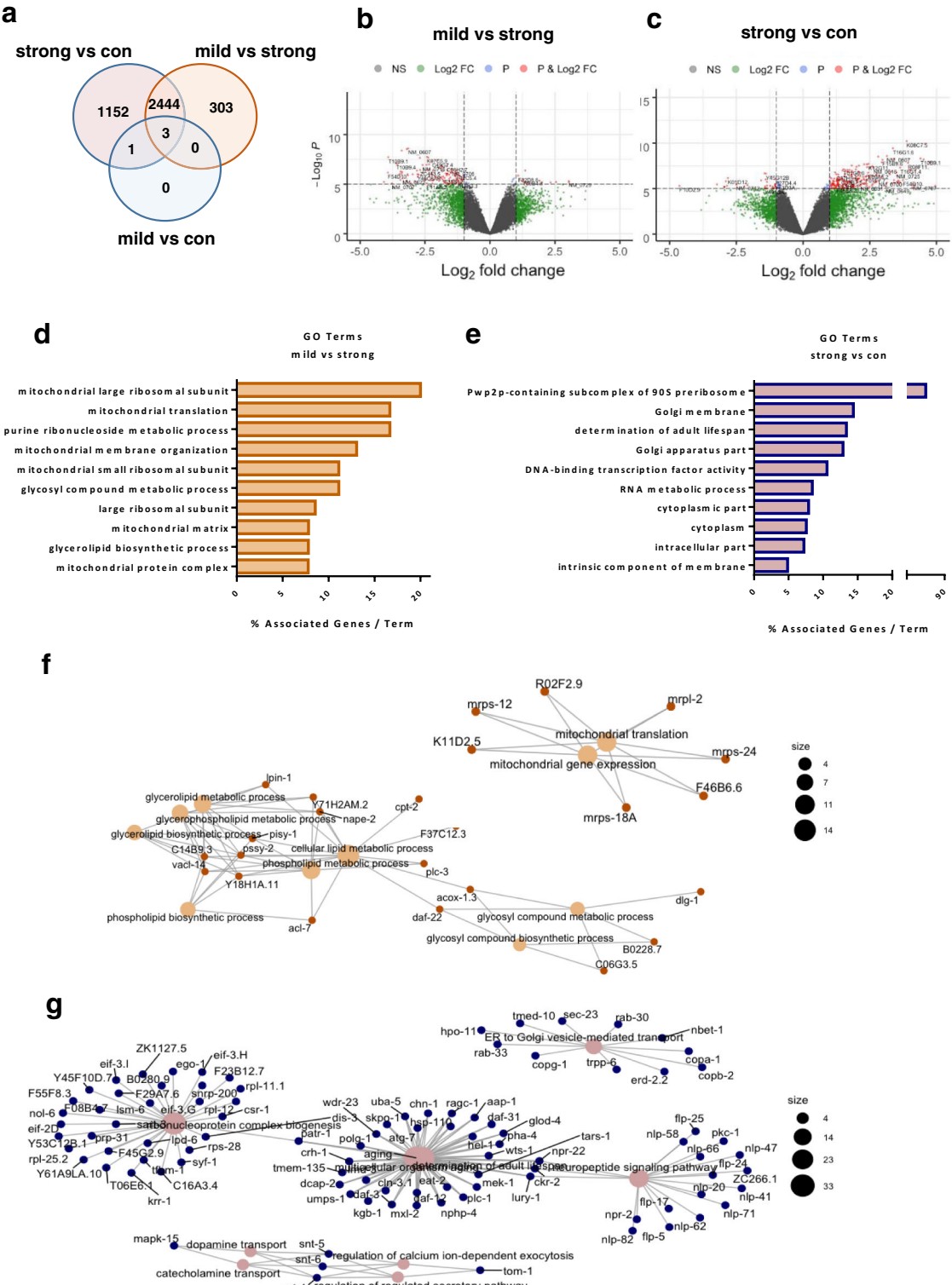

wild-type animals in response to levamisole (Fig. 6b), which is still compatible with exaggerated pre- or post-synaptic activity. This paralysis was however not significantly reverted by lutein pointing towards its protective effect at the presynaptic rather than post-synaptic level or in the synaptic cleft.

The partial effect of lutein on levamisole-induced paralysis could be due to an excessive levamisole concentration and/or activation of postsynaptic receptor compensatory mechanisms or to a concomitant defect in the GABA inhibitory system. To discriminate between these different possibilities, we took

advantage of a complementary assay, which exploits pentylenete-trazole (PTZ)[50], a GABA receptor antagonist. In response to PTZ, neurotransmission mutants display different phenotypes: wild-type worms and mutants with defects in ACh release do not display PTZ sensitivity, due to appropriate regulation of ACh amount and activity; mutants that cannot negatively regulate ACh release, and thus secrete excessive amounts of ACh become completely paralyzed upon PTZ treatment[50]; GABA receptors mutants (e.g. *unc-49*) instead, only display dose-dependent paralysis of the body (but not of the head) resulting in head

**Fig. 4 Mild and strong suppression of *nuo-5* lead to different gene expression profiles. a** Venn diagram of the differentially altered genes identified in the microarray analysis. The transcriptomic profile in control L3 larvae was compared with that of mild or strong (arrested) *nuo-5* RNAi. Volcano plot displaying differential expressed genes between mild vs strong nuo-5 RNAi (**b**) and strong nuo-5 RNAi vs control (**c**). The vertical axis (y-axis) corresponds to the mean expression value of log 10 (*q* value), and the horizontal axis (*x*-axis) displays the log 2-fold change value. Genes left to the vertical dashed line are down-regulated while genes right to the vertical dashed line are up-regulated. The horizontal dashed line separates significant differentially regulated genes (P cut-off $10^{-6}$). Total number of probes = 29317. Gray dots between the two vertical dashed lines denote genes with no significant difference. Plots were designed in R, using Bioconductor. For multiple comparisons adjustments were made with Benjamini–Hochberg (BH) method. **d** GO analysis of the 303 uniquely altered genes between mild vs strong *nuo-5* RNAi and **e** of the 1152 genes uniquely altered between strong *nuo-5* RNAi vs control. **f** Network plot of enriched terms between strong *nuo-5* RNAi vs control and **g** for mild vs strong *nuo-5* RNAi were created in R, using Bioconductor. For statistical analysis Enrichment/Depletion, two-sided hypergeometric test was used followed by correction with Bonferroni step down; only statistically significant GO terms are shown, corrected *p* values < 0.05.

convulsions (head bobs). We found that *nuo-5* depleted worms responded to PTZ with a significantly higher percentage of full-body paralysis, and only a small non-significant fraction of head convulsions; most importantly, lutein treatment significantly suppressed the full body paralysis but not the head convulsions (Fig. 6c). Overall, our paralysis assays: lead to the identification of a pre-synaptic defect induced by severe mitochondrial deficiency; are specifically consistent with the inability to negatively regulate ACh release, while excluding a major defect in GABAergic transmission[51]; indicate a rescuing effect of lutein on the altered negative feedback on ACh release (Supplementary Table 5).

**Synaptic neuroligin expression is increased upon CI deficiency and suppressed by lutein.** We then turned to our microarray results to identify possible pathogenetic targets of lutein. We confirmed the effect of lutein on some *nuo-5* RNAi-induced genes, such as *vps-35*, *nlp-40* and *unc-10* (Supplementary Fig. 11a), highly representative of the two main GO categories (Fig. 5c). We also exploited GFP reporter strains to analyze the expression of different genes specifically involved in synaptic functionality: the expression of *unc-10*, involved in synaptic membrane exocytosis[54], was significantly induced by *nuo-5* RNAi and suppressed by lutein; the expression of *glb-10*, required for synaptic structure and function[55], was reduced by *nuo-5* RNAi and returned to levels similar to control upon lutein treatment; *unc-17*, a gene required in cholinergic neurons to load acetylcholine into synaptic vesicles[56], was increased by *nuo-5* RNAi and reduced to control levels upon lutein treatment (Fig. 6d–f; Supplementary Fig. 11b–d). Moreover, we plotted the expression of genes involved in synaptic vesicle cycle from the microarray and found them to be all significantly altered in *nuo-5* animals compared to control (Fig. 6g). However, when we compared the expression of these genes in *nuo-5* vs *nuo-5* treated with lutein, the only significantly altered transcript was *nlg-1* (Fig. 6h). NLG-1 is a presynaptic cell adhesion protein, neuroligin, which along with postsynaptic neurexin (NRX-1), mediates in *C. elegans* a retrograde signal that inhibits neurotransmitter release at NMJ[57]. Remarkably, we found a conserved and consistent upregulation of different neuroligin (NLGN1-2 and 4) and neurexin (NRX1 and 3) genes in the brain of *Ndufs4* KO mice compared to wild-type animals (Fig. 7a; Supplementary Fig 11e–f). Closer analysis of *nlg-1* interactors from the microarray revealed the expression of two of them (*madd-4* and *dlg-1*) to be significantly changed by *nuo-5* RNAi compared to control and, although not in a significant manner, to be altered upon lutein treatment (Fig. 7b, c). Using a *C. elegans* transgenic reporter strain to quantify the expression of *nlg-1*, we confirmed a significant increase in *nuo-5* depleted animals which went back to wild-type levels upon lutein treatment (Fig. 7d, e). These observations along with the altered behavioral assays, are indicative of an excessive buildup of Ach in the synaptic cleft in the disease model, a defect that is considerably rescued by lutein.

Upregulation of *nlg-1* is required in *C. elegans* to enhance synapses strength and prevent neuronal degeneration in response to oxidative stress[58]. Thus, to assess whether lutein reduced *nlg-1* expression and rescue synaptic functionality possibly through its antioxidant activity we tested the effect of above-mentioned antioxidants in *nuo-5* depleted animals on aldicarb-induced paralysis and *nlg-1* expression. Very interestingly, within different antioxidants, NAC was the only one capable of suppressing aldicarb sensitivity to a similar extent than lutein (Fig. 7f), but contrary to lutein it did not rescue *nlg-1* overexpression (Fig. 7g). These data are in agreement with our cellular and transcriptomic analysis and suggest that lutein may exert its beneficial effect at synapse independently of its antioxidant activity.

**Neuroligin mediates the protective effect of lutein on cholinergic synapses.** We then reasoned that *nlg-1* is increased in *nuo-5* depleted animals to compensate for a mitochondrial-stress-induced synaptic defect, and its downregulation upon lutein treatment likely represents a consequence of its beneficial effects on specific mitochondria-regulated neuronal functions. In this scenario knock-out of *nlg-1* should worsen the deleterious effects of *nuo-5* RNAi and lutein may still retain some beneficial effects. Interestingly, contrary to our expectation *nlg-1* knockout per se significantly ameliorated neuromuscular defects (thrashing, sensitivity to aldicarb and levamisole, as well as full body paralysis in response to PTZ, Fig. 8a–d; Supplementary Fig. 12a–g) induced by *nuo-5* silencing, without any obvious alteration in control animals. Strikingly, a similar protective effect of *nlg-1* knockout was observed against *nuo-5*-induced developmental defects in two different *nlg-1* knock-out alleles, thus excluding unspecific effects due to animals' genetic background (Fig. 8e; Supplementary Fig. 13a). These results imply that induction of *nlg-1*, contrary to the protective effect that has been observed against acute oxidative stress[58], has instead a detrimental effect when upregulated in response to chronic and severe mitochondrial dysfunction. In strong support of a beneficial effect for *nlg-1* suppression, the *nlg-1* knockout strain also clearly masked the beneficial effects of lutein in mitochondria defective animals (Fig. 8a–e; Supplementary Fig. 12a–g).

Of note, in further support of a detrimental effect ascribed to *nlg-1* overexpression upon CI deficiency, two strains over-expressing NLG-1, namely a translational reporter P*nlg-1*::NLG-1::GFP[58] and a *nlg-1*(ok259) knockout strain rescued by a P*nlg-1*::NLG-1 array[59], displayed already severe neurodevelopmental phenotypes when grown on *nuo-5* or *lpd-5* RNAi in the parental generation. Specifically, when the transgenic P*nlg-1*::NLG-1::GFP was fed *nuo-5* or *lpd-5* RNAi we could already observe an increased expression of *nlg-1* in the parental generation (Fig. 8f), which became sterile. Accordingly, also the *nlg-1*(ok259) rescued strain became sterile upon mitochondrial deficiency already in the parental generation (Supplementary Fig. 13a). Most importantly, NLG-1 rescue restored the beneficial effect of lutein in the

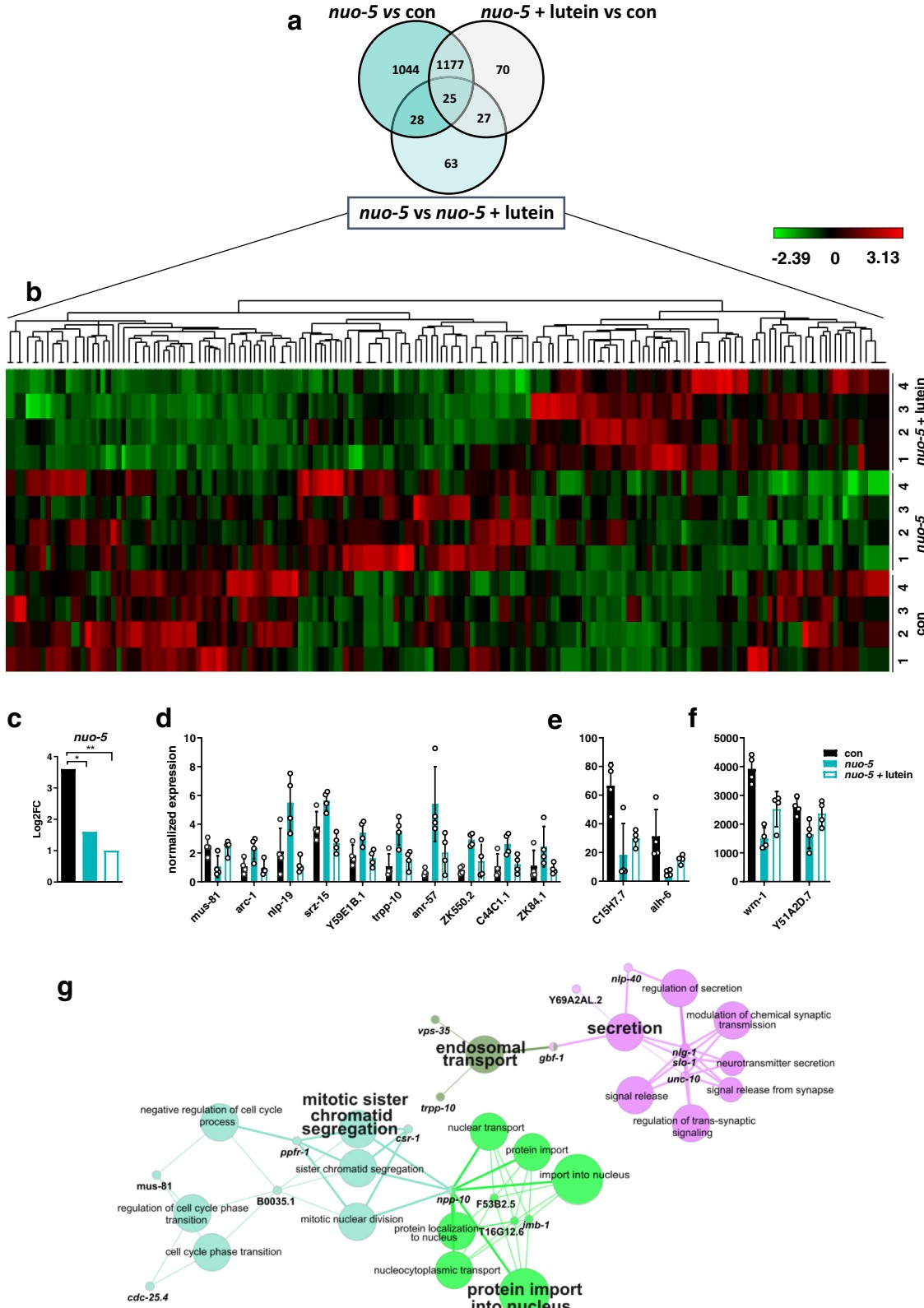

aldicarb sensitivity assay to the degree of the wild-type animals (Fig. 8g) confirming lutein works through suppression of *nlg-1*. Overall, we revealed a pathogenetic role for *nlg-1* overexpression upon mitochondrial dysfunction, which mediates an altered synaptic functionality restored by lutein, thus pointing towards a novel potential targeted therapeutic for the corresponding human pathologies.

## Discussion

In the present work, we developed and thoroughly characterized *C. elegans* models for mitochondriopathies, and focused on CI deficiency associated to *NDUFS1/nuo-5* and *NDUFS4/lpd-5* depletion to carry out a phenotype-based screening in search of disease suppressors. We identified a nutraceutical, lutein, which significantly prevented the neurodevelopmental deficits observed

**Fig. 5 Microarray analysis of worms treated with bacteria expressing empty-vector control, or strong RNAi against *nuo-5* or *nuo-5* treated with lutein.**
**a** Venn diagram obtained using the online tool: http://bioinformatics.psb.ugent.be/webtools/Venn/. **b** Unsupervised hierarchical clustering of the transcripts corrected by treatment of the mutant *nuo-5* with 1 μM lutein. Each of the twelve lines represents a sample, and each of the columns represents a transcript (list of genes is Supplementary Data 5). Expression levels (log2FC) are represented by a heatmap (the scale is included in the top-right quadrant, green represents down-regulation, min log2FC = −2.39; red represents up-regulation, max log2FC = 3.13). The microarray data were obtained with a total of 16 *C. elegans* samples. The analysis included the 143 genes significantly differentially expressed between *nuo-5* vs *nuo-5* + lutein (light blue circle in the Venn diagram in panel **a**). **c** Log2FC (base-2 logarithm of the fold change) of *nuo-5* expression extracted from microarray data; *p* values con vs *nuo-5* p = 0.024, con vs *nuo-5* + lutein p = 0.0013. Gene expression from microarray data showing representative genes among those from the heatmap (Fig. 5b) divided in three groups based on their expression levels: **d** low (<10), **e** middle (between 10 and 100) and **f** high (>100) level. Bar graphs represent means ± SD. **g** Pathways enriched in the microarray *nuo-5* vs *nuo-5* + lutein designed using ClueGO v2.5.

in the two disease models. Importantly, we found that lutein exerts its beneficial effect by rescuing a synaptic defect, which we described in *NDUFS1/nuo-5* depleted animals. We specifically identified overexpression of the *C. elegans* neuroligin homolog, *nlg-1*, as a mediator of the developmental and synaptic defects observed upon severe mitochondrial dysfunction and suppressed by lutein (Supplementary Table 5) which, based on the conserved overexpression in mice, may represent a new relevant pathogenetic and therapeutic target for the human disease. In this regard, it is also very important to note that, at least at the dose used in this study, lutein had no effects in the wild-type animals. This indicates lutein specifically targets pathogenetic mechanisms induced by CI deficiency, with no "side effects" in a non-compromised background, a very encouraging indication for the development of new potential therapeutic strategies.

To date, experimental animal models for CI disorders are scarce (e.g. *Ndufs4* mutant mice)[60], do not fully recapitulate the disease state (e.g. *Surf1* mutant mice)[61] or do not exist (e.g. *Ndufs1*) thus seriously hampering the development of adequate therapeutic strategies, which indeed are not available for these devastating disorders with fatal outcome. As a consequence, studies revolved at investigating CI disorders and potential therapeutic strategies are primarily carried out in patients' fibroblasts or lymphoblasts, which, although certainly useful, are not the most relevant systems to recapitulate diseases of the nervous system[2]. To overcome cell-specific limitation, more recent and exponentially growing approaches (including our ongoing studies) rely on iPS-derived neuronal systems[62]. Although very promising, these structures lack the complexity of the whole organism thus potentially precluding identification of important cell-non-autonomous effects induced by mitochondrial dysfunction, which may have important implication for disease development, progression and therapy. This is supported by our data in the neuronal specific and hypersensitive RNAi backgrounds clearly indicating systemic mitochondrial suppression has stronger and broader repercussion than neuronal-restricted silencing. Therefore, in the first part of this work, to generate animal models for mitochondriopathies, we carried out a customized RNAi screen against genes whose severe deficiency in humans is associated with mitochondrial disorders. While we utilized phenotypic features already described by us and others with some of the targeted genes (e.g. *nuo-2*), such as slow growth and reduced fertility, our targeted approach allows us to screen for additional phenotypes, such as animals' paleness, sickness and defects in the progeny, and to deeply characterize selected clones in more details in terms of survival and neurometabolic features. We identified for half of the clones under study a dose-dependent response to the gene silencing, which mimics the disease progression and is consistent with the mitochondrial threshold effect. Namely, compensatory metabolic reprogramming and signaling pathways may help coping with a mild mitochondrial stress, yet, after a certain threshold of mitochondrial dysfunction is reached, these processes are no longer protective and detrimental features arise, such

as mitochondrial and neurodevelopmental deficits[7,17,19,21]. Very interestingly, we found that animals with mild and severe mitochondrial disruption were in most cases equally long-lived compared to wild-type, indicating that signaling pathways specifying development and lifespan upon mitochondrial stress are uncoupled. Consistent with this scenario, the gene expression profiles in the L3 larvae indicated that remodeling of mitochondrial structure and function underlies the main (developmental) differences observed between mild and strong *nuo-5* suppression. Yet, changes in lifespan-regulatory pathways normally observed in adult upon pro-longevity mitochondrial stress[45,46] are not observed in the L3, while are already visible upon severe mitochondrial suppression. Along with the respiration profiles, these data support previous works indicating that mitochondrial changes are not all relevant for longevity specification but rather correlate with developmental outcomes[21,63,64].

Regardless of lifespan specification, we showed that about 50% reduction in *nuo-5* transcript expression is clearly sufficient to significantly reduce neurometabolic parameters (around 50%) and to induce changes in the expression profile of mitochondrial and metabolic genes, which are very similar to the mitochondrial deficits[2] and the proteomic changes[28] observed in NDUFS1 patients' fibroblasts. Severe suppression of different mitochondrial diseases-associated genes induced very reproducible *C. elegans* phenotypes (developmental arrest or reduced body size), which were essential to carry out our suppressor screen. Yet, consistent with the heterogeneity of signs and symptoms of these mitochondrial disorders[28,65], we also observed variability in the extent of different phenotypic outcomes e.g. locomotion and chemosensory deficits, mitochondrial respiratory parameters, ability of compounds to rescue the developmental arrest. We initially assumed the movement defect simply arises at a later age, as previously described[24], but we could actually uncover it under increased energetic demand. Interestingly, this behavior is similar to the perceived fatigue often observed in patients with mitochondrial disorders[25] and given the suppressive effect of lutein, it provides us with a very sensitive and quick assay for future suppressor screening. It will be very important to further investigate whether depletion of other CI subunits or of other mitochondriopathies associated genes increases sensitivity to metabolic or environmental/dietary stressors as these factors may influence mitochondrial activity and have devastating consequence in patients[66,67]. Interestingly, we also found a defect in animals' pharyngeal pumping, a very relevant phenotype considering the similarities between *C. elegans* pharynx and the human heart, and the fact that patients with mitochondrial disorders often present with, and die for, cardiac dysfunction.

At the cellular level, we observed increased ROS production in both *C. elegans* and patients' fibroblasts, accompanied by increased expression of mitochondrial stress response genes (i.e. *gst-4, hlh-30, dct-1*, and *hsp-6*) and increased sensitivity to oxidative stress in *nuo-5*- and *lpd-5* depleted animals. Yet, at least at this early stage we have looked at, oxidative stress and damage do

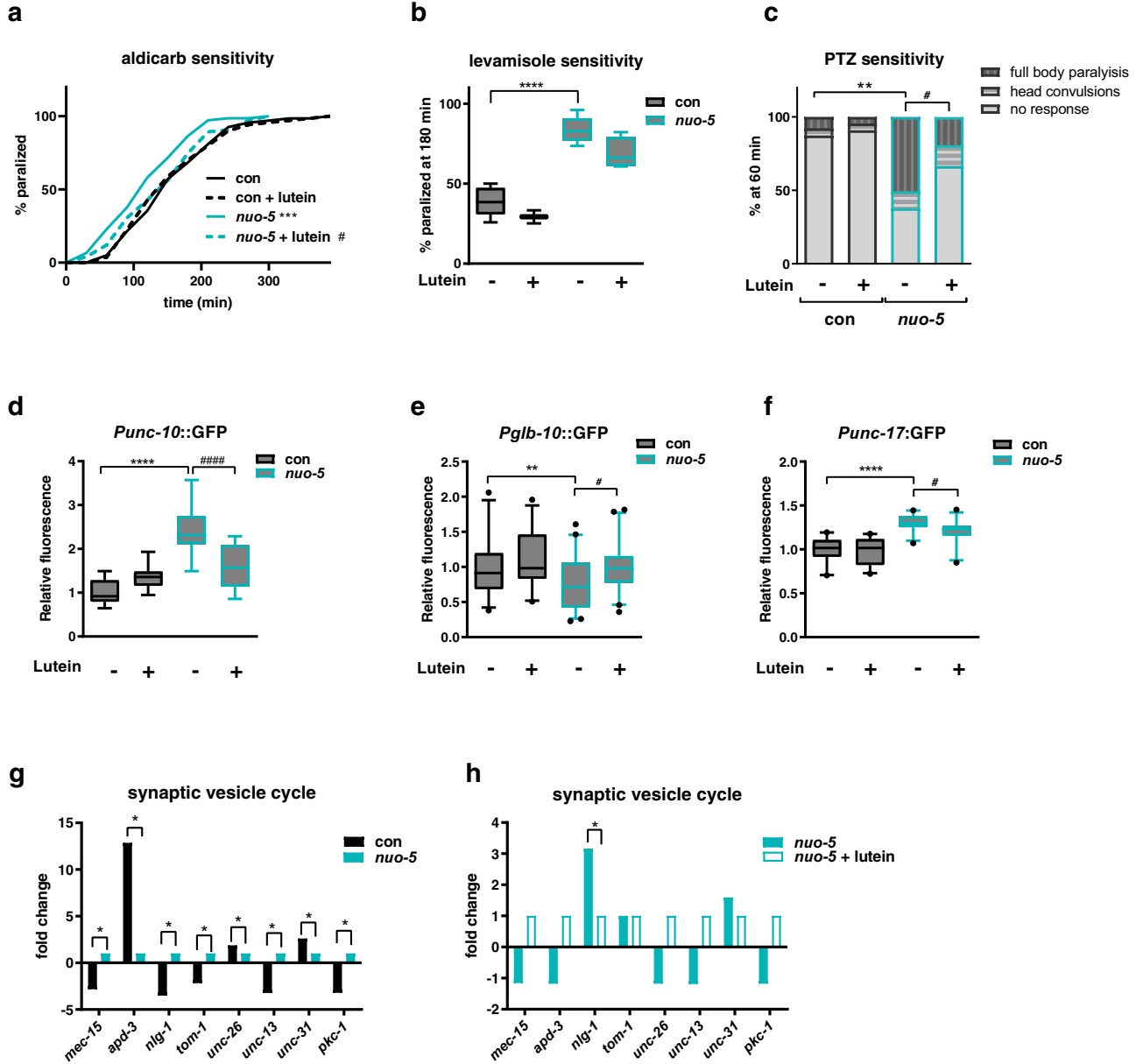

**Fig. 6 Lutein suppresses an *nlg-1*-dependent synaptic defect in *nuo-5*-depleted animals.** Worms in all panels were fed bacteria transformed either with empty-vector (con, black bars) or with vector expressing dsRNA against *nuo-5* (light blue bars) and left untreated or treated with 1 μM Lutein. **a** Aldicarb-induced paralysis curves on plates containing 0.5 mM aldicarb. Log-rank test: con vs *nuo-5* p = 0.0004, *nuo-5* vs *nuo-5* + lutein p = 0.018. n ≥ 75 over three biologically independent experiments. **b** Percentage of animals paralyzed on plates containing 50 μM Levamisole. Box plots indicate median (middle line), 25th, 75th percentile (box) and 5th and 95th percentile (whiskers) as well as outliers (single points). n ≥ 60 over four biologically independent experiments. con vs *nuo-5* p < 0.0001. **c** Bars indicate percentage of animals with the indicated phenotypes after 1 h of exposure to 2.5 mg/ml PTZ. n = 60 over four biologically independent experiments. Asterisks denote the differences only of the full body-paralysis phenotype. con vs *nuo-5* p = 0.0085, *nuo-5* + lutein p = 0.046. **d–f** Quantification of GFP expression in the indicated transgenic strains. Box plots indicate median (middle line), 25th, 75th percentile (box) and 5th and 95th percentile (whiskers) as well as outliers (single points). n ≥ 45 over three biologically independent experiments. Asterisks (*) denote significant differences vs control, # denote differences among conditions. p values in **d**: ****p values < 0.0001, p < 0.0001. p values in **e**: con vs *nuo-5* p = 0.0066, *nuo-5* vs *nuo-5* + lutein p = 0.0112. p values in **f**: con vs *nuo-5* p < 0.0001, *nuo-5* vs *nuo-5* + lutein p = 0.0474. Two-way ANOVA followed by Tukey's multiple-comparisons test, unless otherwise specified. Data for all panels are provided as a Source Data file. **g, h** Fold change of genes involved in synaptic vesicle cycle expression (from microarray data).

not seem to play a major role in eliciting or suppressing animals' phenotypes. In fact, proton leak as well as lipid and amide oxidation were actually reduced and a classical oxidative stress response gene (p38MAPK/PMK-1) was not activated in the disease models. Furthermore, while lutein significantly suppressed neurodevelopmental deficits, as well as the induction of *hlh-30* in

the two disease models, it neither suppressed the induction of *gst-4* or *hsp-6* nor modulated animals' sensitivity to oxidative stress. Interestingly, *dct-1* expression was regulated in opposite direction by lutein in the two disease models, perhaps indicating that an optimal level of mitophagy must be achieved for proper mitochondria turnover. Thus, *hlh-30*/TFEB-regulated autophagy

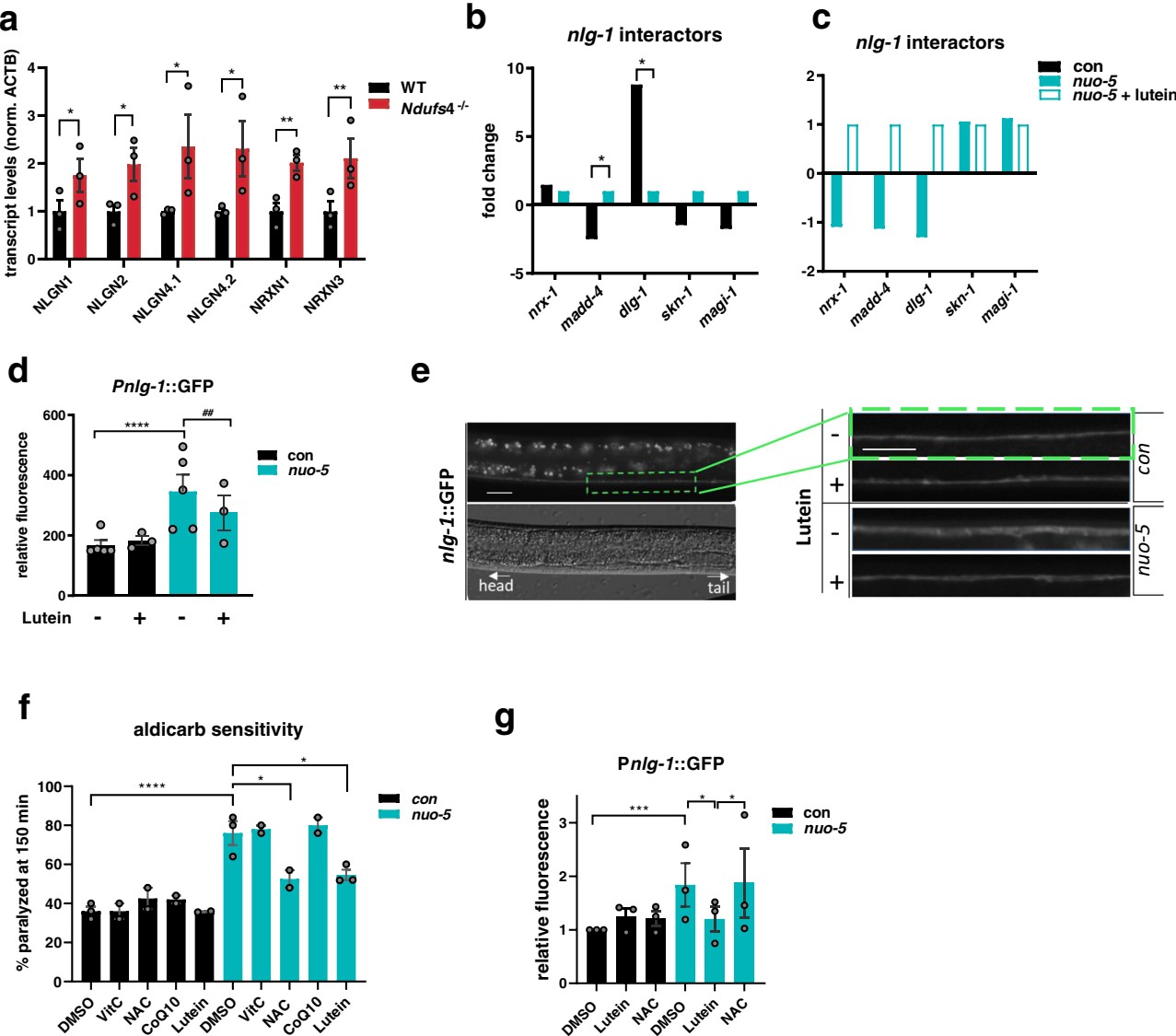

**Fig. 7 Upregulation of neuroligin and neurexin genes upon *nuo-5* is conserved across species and the defect is rescued by lutein but no other antioxidants. a** Expression levels of neuroligin (NLGN1-2 and 4) and neurexin (NRX1 and 3) transcripts in brains of NDUFS4−/− mice compared to healthy animals normalized to ACTB, assessed by qPCR. Three biologically independent experiments, n = 3. NLGN1 p = 0.0245, NLGN2 p = 0.019, NLGN4.1 p = 0.0179, NLGN4.2 p = 0.0104, NRXN1 p = 0.0013, NRXN3 p = 0.0068. **b**, **c** Fold change of genes described as *nlg-1* interactors (from microarray data). **d** Quantification of GFP expression of p*nlg-1* transgenic strain. Box plots indicate median (middle line), 25th, 75th percentile (box) and 5th and 95th percentile (whiskers) as well as outliers (single points). n ≥ 47 over three biologically independent experiments. con vs *nuo-5* p < 0.0001, *nuo-5* vs *nuo-5* + lutein p = 0.0023. **e** Representative pictures of p*nlg-1*::GFP in ventral nerve cord. Scale bar 10 µm. Three biologically independent experiments with similar results. **f** Sensitivity to Aldicarb is shown as percentage of paralyzed animals after 150 minutes on plates containing 0.5 mM Aldicarb. Bar graphs represent means ± SEM and corresponding data points. n ≥ 50 in two biologically independent experiments for VitC, NAC, CoQ10 and three biologically independent experiments for DMSO and lutein. con DMSO vs *nuo-5* DMSO p < 0.0001, *nuo-5* DMSO vs *nuo-5* NAC p = 0.029, *nuo-5* DMSO vs *nuo-5* lutein p = 0.024. **g** Quantification of GFP expression in P*nlg-1*::gfp transgenic strain. Box plots indicate median (middle line), 25th, 75th percentile (box) and 5th and 95th percentile (whiskers) as well as outliers (single points). n ≥ 36 over three biologically independent experiments. con vs *nuo-5* DMSO p = 0.0004, *nuo-5* DMSO vs *nuo-5* lutein p = 0.025, *nuo-5* lutein vs *nuo-5* NAC p = 0.0172. One-way ANOVA, bar graphs represent means ± SEM, asterisks (*) denote significant differences vs control, # denote differences among conditions. Data for all panels are provided as a Source Data file.

and *dct-1*/Bnip3-regulated mitophagy may represent protective mechanisms against the mitochondrial defects or may induce deleterious effects if over-activated in the disease. Regardless, fine-tuning of these processes, which have been already implicated in the pathogenesis of different neuromuscular disorders, may be seen as part of a cytoprotective effect promoted by lutein.

Remarkably, through a combination of genetic and behavioral assays we were able to uncover an acetylcholine presynaptic alteration upon severe mitochondrial dysfunction (most likely an

impairment in the ability to negatively regulate ACh release at the NMJ), which was prevented by lutein treatment. Of note, both mitochondria and autophagy play critical role in shaping synaptic structure and function[68,69] and our microarray data pointed towards genes regulating neurotransmitter secretion and endosomal transport as two out of four main pathways affected by lutein in the disease model. It will be thus very interesting to untangle how CI deficiency provokes synaptic alteration and whether autophagy and/or mitophagy, which we found to

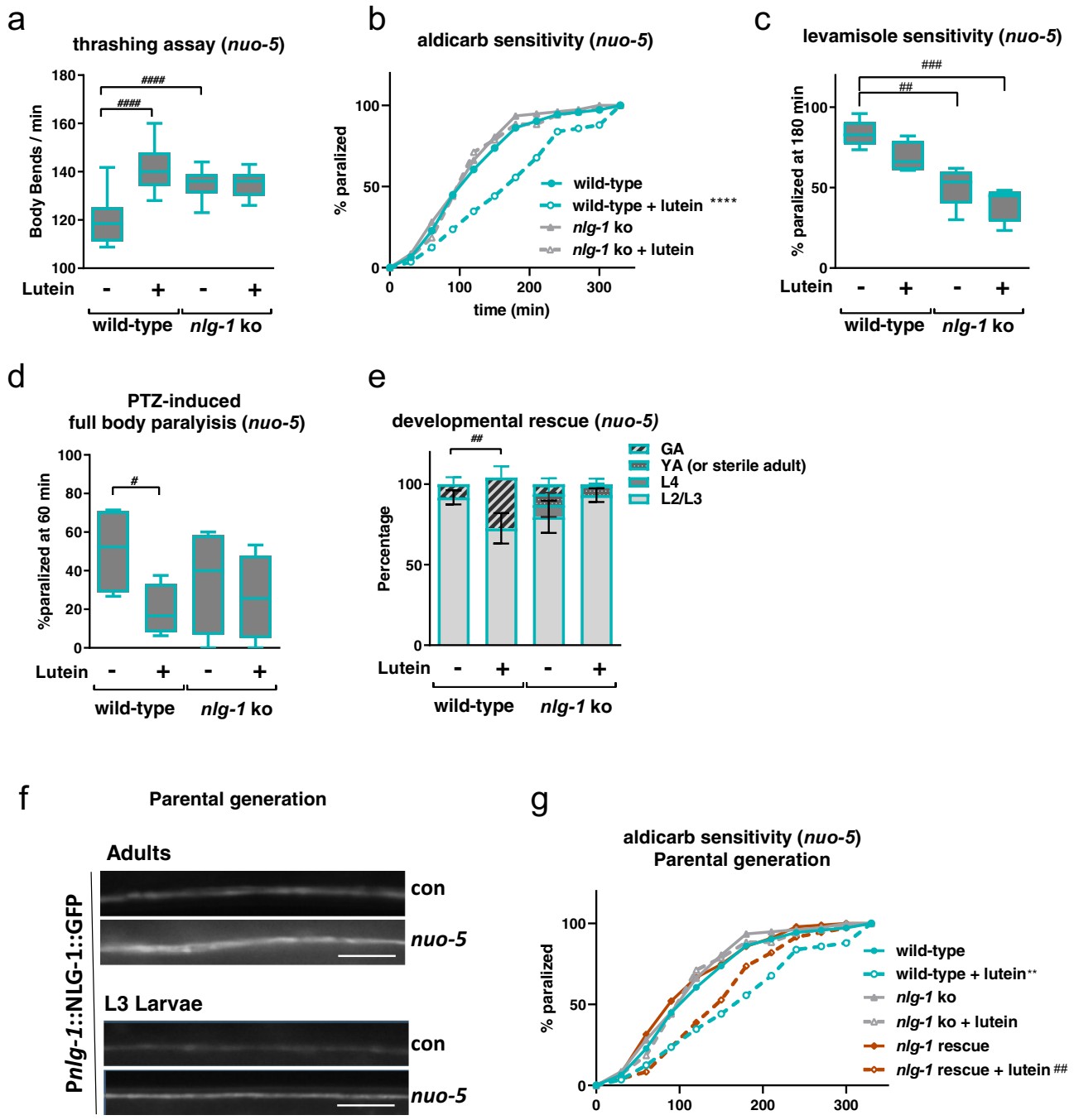

**Fig. 8 Neuroligin mediates the protective effect of lutein on cholinergic synapses. a** Frequency of body bends in liquid media ("thrashing") were quantified manually for one minute. $n \geq 45$ over three biologically independent experiments. $p < 0.0001$. **b** Aldicarb-induced paralysis curves on plates containing 0.5 mM aldicarb. Log-rank test: wild-type vs wild-type + lutein $p = 4.40^{E-07}$. $n \geq 89$ over three biologically independent experiments. **c** Percentage of animals paralyzed after 180 min on plates containing 50 μM Levamisole. Box plots indicate median (middle line), 25th, 75th percentile (box) and 5th and 95th percentile (whiskers) as well as outliers (single points). $n = 120$ over four biologically independent experiments. wild-type vs nlg-1 ko $p = 0.0012$, wild-type vs nlg-1 ko + lutein $p = 0.0001$. **d** Bars indicate the percentage of animals showing full-body paralysis after one hour of exposure to 2.5 mg/ml PTZ. $n = 60$ over four biologically independent experiments. wild-type vs wild-type + lutein $p = 0.0029$. **e** Bars indicate the percentage of different developmental stages in wild-type and nlg-1 mutants after 6 days from egg lay. Asterisks denote differences between the percentages of animals that reached the gravid adult (GA) stage. $n \geq 120$ over four biologically independent experiments, two-way ANOVA, followed by Tukey's multiple-comparisons test. wild-type vs wild-type + lutein $p = 0.0049$. **f** Representative pictures of $p_{nlg-1}$::NLG-1::GFP in ventral nerve cord of adult and L3 larvae in the parental generation. Scale bar 10 μm. Three biologically independent experiments with similar results. **g** Aldicarb sensitivity assay in the nlg-1(ok259) rescued by a $p_{nlg-1}$::NLG-1 transgene, aldicarb-induced paralysis curves, plates containing 1 mM aldicarb. Log-rank test: wild-type vs wild-type + lutein $p = 0.004$, nlg-1 rescue vs nlg-1 rescue + lutein $p = 0.0064$. $n \geq 75$ over three biologically independent experiments. The assay was carried out using adult worms of the parental generation. Data for all panels are provided as a Source Data file.

mediate mild mitochondrial stress extension of lifespan[35,70], also play a role in this pathological context.

Intriguingly, we showed that within different synaptic regulatory genes affected by *nuo-5* depletion, lutein treatment consistently and significantly abolished the upregulation of neuroligin (*nlg-1*), a cell adhesion synaptic protein that along with its partner protein neurexin, mediate a retrograde synaptic signaling that inhibits neurotransmitter release[57,59]. Interestingly, abnormal expression of extracellular adhesion molecules has been observed in NDUFS1 fibroblast[28], yet, not surprisingly, we could not find any specific change in neuroligin expression in the NDUFS1 fibroblasts (Supplementary Fig. 11g). Instead, most importantly, we found different neuroligin and neurexin genes also significantly upregulated in the brain of *Ndufs4* knockout mice, suggesting abnormal synaptic neuroligin expression as a common pathogenetic denominator of different CI-linked disorders. In support of a detrimental effect of neuroligin overexpression, *nlg-1* knockout rescued the neuronal deficits and, strikingly, also partially the developmental defect, induced by mitochondrial dysfunction, masking the beneficial effects of lutein. This is opposite to the protective role previously described in *C. elegans* neurons, where induction of *nlg-1* by SKN-1/Nrf2 (which is activated by lutein in mammals[71]) confers resistance against acute oxidative stress[58]. Our data clearly suggest *nlg-1* may play opposite effects depending on the context and imply lutein suppresses its expression likely through ROS-independent mechanisms modulated downstream to mitochondria. Accordingly, although *nuo-5* and *lpd-5* depleted animals were more sensitive to oxidative stress, lutein neither affected this response, nor suppressed *nuo-5* RNAi-induction of *gst-4* (a well-established SKN-1 target in response to oxidative stress) or of other mitochondrial stress response genes. In support of ROS-independent regulation of *nlg-1* in this context, whilst interrogating the effects of different antioxidants known to rescue deleterious phenotypes in other CI disease models (i.e. NAC, VitC and CoQ10[16]), we found that only NAC improves the synaptic functionality of *nuo-5* depleted animals yet without suppressing *nlg-1* overexpression. It will be interesting to further investigate how abnormal expression of neuroligin impacts on animals' neuronal function and development in response to mitochondrial deficiency.

Mitochondria regulate synaptic activity in many ways and severe mitochondrial dysfunction is therefore expected to have broader repercussion on synaptic regulatory proteins[68]. Increased expression of NLG-1 may thus represent a neuronal attempt to compensate for compromised synaptic signaling induced by the mitochondrial deficiency. However, rather than protective, this would lead to a pathological synaptic excitation, which, as supported by our transcriptomic and behavioral (aldicarb, levamisole and PTZ sensitivity) data, was in fact rescued by lutein suppression of NLG-1. Chronic exposure to lutein in the diseased worms could thus activate NLG-1-regulated compensatory mechanisms that rescued their paralysis defects. Interestingly, NLG-1 contains an extracellular cholinesterase-like domain that forms the interaction with the neurexin domain[72], and lutein was shown to have acetylcholinesterase (AChE) inhibitory activity[73]. Whether lutein neuroprotection upon CI deficiency is mediated by direct inhibition of synaptic AChE and/or NLG-1 activity (in turn suppressing compensatory *nlg-1* overexpression), is an attractive possibility that requires further investigation. It is worth noting that Alzheimer's disease (AD) patients were reported to have significantly less macular pigments (comprising lutein, zeaxanthin and meso-zeaxanthin) and lower serum concentrations of lutein and zeaxanthin compared to control subjects[74] and that AChE inhibitors are the standard therapy for AD[75]. Moreover lutein was shown to have beneficial effects in AD patients[43] (who interestingly also display neuroligin-mediated synaptic

dysfunction[76]), as well as against other neuropathological conditions such as diabetic retinopathy[77], age-related macular degeneration[42], and Parkinson disease[78], primarily due to its antioxidant, anti-apoptotic and mitochondrial protective properties. We provide the proof of principle for considering lutein as an easily accessible therapeutic for mitochondrial diseases associated with CI deficiency, which to date present with no cure. Lutein can be naturally ingested either through the diet (dark leave vegetables) or as a dietary supplement or as an ingredient in nutraceutical foods. However, due to its highly hydrophobic nature different strategies are being developed to improve its bioavailability[73,79,80], which can therefore offer more effective therapeutic strategies. Finally, besides lutein, our suppressor screen identified few other compounds that partially rescued the developmental arrest of CI deficient animals and it will be interesting to assess them alone or in combination with lutein in the different neuronal assays.

In summary, in this work we established *C. elegans* models for devastating neurodevelopmental disorders, which to date present with no cure, and focused our attention on the CI deficiency model associated with NDUFS1 depletion, to our knowledge the first and only animal model available for this disease. This simple (yet multicellular) model organism, with its reduced nervous system complexity and powerful neurobehaviors represents a great opportunity to identify early pathomechanistics defects causally involved in the disease. Our results support the growing body of evidence indicating that live animal screens are ideal tools for drug discovery strategies designed to identify suppressors of specific and complex disease-associated phenotypes[81–83]. Most importantly, although validation in mammalian systems is required to encourage translational studies, we provided the rationale for suggesting lutein as a therapeutic to rescue a neuroligin-mediated synaptic dysfunction upon CI deficiency.

## Methods

**Ethical statement.** Our research complies with all relevant ethical regulations. No ethical approvals are required for laboratory studies using the nematode *C. elegans*. The use of patient-derived cell lines (fibroblasts) was approved by the local ethic committee of the Heinrich Heine University of Düsseldorf (study number #4272). Written informed consent was obtained from the parents.

### *C. elegans* methods

*C. elegans* *strains and maintenance*. We employed standard nematode culture conditions[24]. All strains were maintained at 20 °C on Nematode Growth Media agar supplemented with *Escherichia coli* (OP50 or transformed HT115), unless otherwise indicated. Strains used in this work are listed in Supplementary Methods (Supplementary Table 7). All our neuromuscular and mitochondrial assays are carried out on L3 larvae unless otherwise specified.

*RNAi feeding*. The following dsRNA transformed bacteria for feeding were derived from the Ahringer *C. elegans* RNAi library[84]: *nuo-1* (C09H10.3), T20H4.5, F53F4.10, *cco-1* (F26E4.9), *nuo-5* (Y45G12B.1), *nuo-2* (T10E9.7), *spg-7* (Y47G6A.10), *lpd-5* (ZK973.10). For additional RNAi used in the phenotypic screening described in the text refer to Table 1 and Supplementary Table 1. All dsRNA bacterial clones (sequence validated), were grown to a concentration of 0.9 OD and diluted 1/10 or 1/50 with empty vector expressing bacteria.

*Lifespan assay*. Lifespan and Statistical Survival analysis using synchronous populations of at least 60 animals per strain. Survival curves and statistical analyses were carried out as described in details in Supplementary Methods. See also Supplementary Table 2 and Supplementary Data 1 for the summary of lifespan statistics.

*Chemotaxis assay*. Animal's sensory neurons functionality was assessed by quantifying attraction or repulsion to different concentration of chemicals as described as in Supplementary Methods. Compounds used in chemotaxis assays were all acquired from Sigma-Aldrich: ammonium acetate (A1542), was diluted in distillated water and 2-Methyl-3-(methylthio)pyrazine (545791), 1-Butanol (B7906), Benzaldehyde (418099), all diluted in Ethanol 99.8%.

*Body bend*. Locomotion was assessed by counting the number of body bends (changes in the body bend at the mid-body point per minute), per minute for each worm on solid agar plates with no bacteria. One bend was counted every time the mid-body reached a maximum bend in the opposite direction from the bend last counted. Body bends were checked in at least 20 single worms in 2 or 3 independent biological trials.

*Thrashing assay*. To further define neuro-muscular differences on an individual level, we analyzed the swimming behavior (also known as thrashing)[85] of *nuo-5* animals. Age-synchronized worms were transferred to a fresh unseeded plate at room temperature to remove leftover bacteria; then individual worms were transferred to an unseeded small NGM plate filled with 1 mL of S-Basal buffer. Worms were allowed to acclimatize for 5 minutes, and the number of completed thrashes per minute was counted using a hand counter. This was performed on 15 individual worms per replicate in 2 or 3 independent biological trials.

*Pharyngeal pumping*. Pharyngeal pumping was measured using the ScreenChip™ System (InVivo Biosystems). This platform allows to record the voltage changes caused by the contraction of the pharynx in real time, producing the so-called EPG[86]. The following parameters were measured: pump frequency, pump waveform shape, spike amplitude ratios, pump duration, IPI and R to E ratio. (Supplementary Fig. 13b shows a representative EPG with details of each parameter). Nematodes were washed off the plates with M9 and collected in a reaction tube as adults in the parental generation instead of L3. This technical adjustment was necessary since the size of the cartridges are to date not compatible with the size of our L3 larvae (the channels in the SC20 are too big while S30 are too small). Then, the worms were washed three times with M9 and incubated in a 10 mM serotonin solution (Sigma Aldrich, 14927) for 30 min. Worms were loaded on the ScreenChip SC40 with a syringe (0.01–1 ml). The EPG of single worms was recorded for a duration of about 2 min. Only worms that showed pumping activity were recorded, while those with no pumping activity were discarded. The software programs NemAcquire 2.1, and NemAnalysis 0.2 (https://invivobiosystems.com/product-category/instruments/screenchip-system-software/) were used for recording and analysis respectively.

*Aldicarb-induced paralysis*. The assay was carried out following protocols previously described by others[48]. Briefly, the animals were all L2/L3 larvae on one day of the assay; they were grown on RNAi plates before the beginning of the assay when 25/30 worms were transferred on a small spot (4 μl) of OP50 *E. coli* in the center of the agar plate. The small spot concentrates the animals into a small single field of view, thus making it easier for careful examination of all animals on a plate. We used plates containing either 0.5 mM or 1 mM final aldicarb (Sigma-Aldrich, 33386) concentration in the NGM. After placing the animals on to an Aldicarb plate they were left at RT. Every 30 min, for a total of 5.5 h, the worms were assayed for paralysis. We define a worm as paralyzed by the absence of movement when prodded three times with a platinum wire on the head and tail.

*Levamisole-induced paralysis*. The assay was carried out as previously described by others[49]. The assay was performed similarly to the aldicarb with the only difference that the paralysis was scored every 15 min for 3 h. We used 50 μM final concentration. The levamisole (Sigma Aldrich 31742) stock solution (100 mM in $_{dd}H_2O$) was added to the liquid NGM.

*(PTZ)-induced paralysis*. PTZ sensitivity assay was executed as previously described by others[50]. Briefly, 75 μl of a 100 mg/ml PTZ (Sigma Aldrich P6500) stock solution in $_{dd}H_2O$ was added to plates containing 3 mL of solid NGM, making a final concentration of 2.5 mg/ml. The PTZ plates were allowed to dry for roughly 60–120 min at RT with the lids open. The assay was carried out as the aldicarb assay, with the difference that the animals were scored for head convulsions and full-body paralysis after 30 min and at one hour. The *C. elegans* strain *unc-49*(e407) was used as positive control. This GABAergic mutant shows significantly more anterior convulsion and full-body paralysis compared to wild-type animals (Supplementary Fig. 12e–f and as described in[87]).

*Mitochondrial respiration*. Basal OCR, maximal respiratory capacity, spare respiratory capacity, ATP coupled respiration and proton leak, have been assessed as described in details in Supplementary Methods; all measurements were performed with synchronized L2-L3 *C. elegans*.

*ATP levels*. The intracellular ATP levels were assessed quantifying whole animal luminescence in the *C. elegans* strain PE255 as described in details in Supplementary Methods[88].

*Mitochondrial DNA copy number, mitochondrial DNA, and nuclear DNA damage*. nDNA and mtDNA damage have been evaluated using a quantitative PCR (QPCR)-based method as described in details in Supplementary Methods[89].

*Reactive oxygen species*. Synchronous L3 stage worm populations of each strain were grown on NGM plates spread with RNAi bacteria and 10 μM MitoSOX Red

(Thermo Fisher M36008). After 24-hour incubation, nematodes were transferred to fresh NGM RNAi plates for one hour to clear their guts of residual dye. Living nematodes were paralyzed in situ by directly adding 1 mg/ml levamisole to NGM agar plates. Photographs were taken immediately in the dark using a CY3 fluorescence filter. Subsequently, the terminal pharyngeal bulb was manually circled to quantify the region's mean intensity using ImageJ software (http://rsbweb.nih.gov/ij/).

*Phenotype-based suppressor screening*. The screen was run in 12-well plates. Each well contained 2 ml solid NGM + 50 μL UV killed bacteria lawn + 50 μl drug solution in final 0.25% DMSO (Sigma Aldrich D8418). In each plate 3 wells were used for RNAi without any compound (0.25% DMSO as control) to check the efficacy of the RNAi on the vehicle; and 3 wells for pL4440 RNAi to have a negative control of animal development on the empty vector (Supplementary Fig. 5B). The screening was repeated for each compound between 1 (in case they gave 0 rescue in the first trial) and 5 times (in case we observed a rescue of the arrested development). Approximately 40 eggs were placed in each well and their development was observed every day until the animals became gravid adults (for a maximum of 6 days).

*Quantification of gene expression through fluorescent transgene reporters*. The effect of *nuo-5* and *lpd-5* RNAi and/or lutein (Sigma Aldrich PHR1699) treatment on the induction of different transgenic strains was investigated on synchronized population of L3 worms (unless otherwise indicated). The nematodes were placed in a 14 μl S-Basal drop on a microscope glass slide, anesthetized with NaN₃ 10 mM, covered with a cover slide and immediately imaged. Pictures were acquired with an Imager2 Zeiss fluorescence microscope, magnification tenfold and ZEN 2 (blue edition) software. In each experiment a minimum of 15–20 animals per condition were used. Details for each fluorescent strain are described in Supplementary Methods.

*RNA extraction and microarray*. Total RNA was extracted as described in details in Supplementary Methods. Quality control, hybridization and Affymetrix *C. elegans* Gene 1.0 ST Microarrays scanning were performed at the Center for Biological and Medical Research, Facility of the Heinrich Heine University of Düsseldorf. Results in Fig. 4 were obtained by raw data (Affymetrix.cel-files) preprocessing and analysis using R version 3.5.1 (2018-07-02). The data were RMA normalized by the oligo package and annotated using pd.elegene.1.0.st package[90]. The list of DEG was obtained using LIMMA (Linear Models for Microarray Data) package[91] with False Discovery Rate (FDR) using a cut-off of adjusted *p* value < 0.05, controlled by Benjamini–Hochberg method. The list of unfiltered DEG was plot as enhanced volcano with cut-off 1 for log2FC and $10^{-6}$ for *p* value (Fig. 4b, c). Results in Fig. 5 were obtained analyzing the data with Partek Flow (build 7.0.18.1116). The data were RMA normalized and annotated using the STAR module. The list of DEG was obtained by ANOVA filtering with FDR correction using the GSA module with a cut-off of adjusted *p* value < 0.05. The list of DEG between *nuo-5* and *nuo-5* lutein treated worms was used to perform unsupervised Euclidian hierarchical clustering. The fold changes are represented as log2FC (base-2 logarithm of the fold change). Venn diagrams (Figs. 4a and 5a) and pathways representative for a selected list of genes were visualized as network plot of enriched terms in Fig. 5c was created with ClueGO v2.5[92], and statistical test used was enrichment/depletion (Two-sided hypergeometric test). The correction method used was Bonferroni step down. The network plots of enriched terms in Fig. 4f, g and Supplementary Fig. 9 were created in R, using Bioconductor[93]. Specifically, cnetplot function was used to extract the complex association[94]. R version 3.5.1 (2018-07-02), clusterProfiler_3.8.1, enrichplot_1.0.2, Bioconductor version 3.8.

*RNA extraction and quantitative RT-PCR*. Total RNA extraction was performed as described for the microarray. Three biological replicas were collected to extract the total RNA and the cDNA was synthetized using GoScript™ Reverse Transcription Mix (Promega, A2790). Primer pairs used for the qPCR are listed in Supplementary Table 6.

*Synchrotron Fourier transform infrared microspectroscopy (SR-μFTIR)*. Eggs were placed on agar plates (with or without lutein, Sigma Aldrich PHR1699) and worms collected at L3 stage (stored at 20 °C). After washing three times with MilliQ water, worms were transferred to CaF₂ windows (Crystran, CAFP8-2) and dried in vacuum condition. μFTIR experiments were performed at the MIRAS beamline at ALBA synchrotron, using a Hyperion 3000 Microscope coupled to a Vertex 70 spectrometer (Brucker) equipped with ×36 magnification objective. The measuring range was 6504–000 $cm^{-1}$ and the spectra collection was done in transmission mode at 4 $cm^{-1}$ spectral resolution. For single point measurements, a mercury cadmium telluride detector was used with an aperture dimension of 10 μm × 10 μm and synchrotron light was used co-adding from 128–256 scans. Background spectra were collected from a clean area of each CaF₂ window. A focal plane array detector was used for imaging measurement using the internal source co-adding 256 scans. FTIR data were analyzed using OPUS 7.5 (Brucker), Unscrambler 10.5 and Matlab R2010b (Mathworks). Spectra exhibiting low signal to noise ratio or saturation were deleted. Remaining spectra were corrected for

Resonant Mie Scattering by an open-source algorithm provided by Gardner group[95]. Matrigel (Corning, Matrigel Phenol-free 356237).

Matrigel (Corning, Matrigel Phenol-free 356237) was used as reference spectrum, 10 iterations, a scattering particle diameter between 2–18 μm and a refractive index 1.1-1.5. A second-order Savitsky-Golay derivative with 9 smoothing points and a polynomial order of two was applied to the corrected data to eliminate baseline effects and improve peak resolution. Lipid oxidation, amide oxidation and lutein ratio were addressed by calculating the absorption ratios for each spectrum, $v_s$(C=O) 1745 cm$^{-1}$ /$v_{as}$ (CH$_2$) 2921 cm$^{-1}$, $v_{as}$ (C=O) 1745 cm$^{-1}$/$v_s$(Amide) 1654 cm$^{-1}$ and v(Lutein) 1515 cm$^{-1}$/$v_{as}$ (CH$_2$) 2921 cm$^{-1}$.

### Mouse samples methods

*Quantitative RT-PCR.* cDNA from mice brain (used here for qRT-PCR analysis) was prepared from mice tissue' samples which were previously collected[96,97]. Tissue RNA was extracted with TriReagent (Sigma, T9424) according to supplier instructions. RNA concentration and quality were determined using Nanodrop (PeqLab) and cDNA was synthesized with the iScript cDNA synthesis kit (Bio-Rad, 1708891) following manufacturer's protocol. Each 9 μl reaction for q-RT-PCR was made of 4 μl diluted cDNA, 0.25 μl of each primer (from 25 μM stock) and 4.5 μl of Luna Universal Master Mix (New England Biolabs, M3003S). The q-RT-PCR reactions were run on the QuantStudio 6 Flex Real-Time PCR system (Applied Biosystems) using the BioRad iQ5.2 Software. qPCR results were analyzed using the DDCt method relative to the mean of housekeeping genes (Hprt, Gapdh and ActB). Each biological data point represents the average of two technical triplicates. Primer pairs used are listed in Supplementary Table 6.

### Human cells methods

*Cells and culture conditions.* Primary human skin fibroblasts (NHDF-neo; Lonza, CC-2509) were used as control cell line. Cell culture model systems of NDUFS4 and NDUFS1 deficiency included a primary human skin fibroblast cell line (not authenticated) from a patient with a homozygous nonsense mutation c.20C>G; p.(Ser7*) in NDUFS4 (NM_002495.2) and a primary human skin fibroblast cell line from a patient with a compound-heterozygous mutation c.1855G>A; c.1669C>T in NDUFS1. Cell lines were previously generated[2] and are available from the authors upon reasonable request. All fibroblasts cell lines were cultured in DMEM with GlutaMAX™-I (Gibco by life technologies) supplemented with 10% FBS and 1% penicillin–streptomycin (5000 U/ml). Medium was replaced every 2–3 days and cells were split when the desired confluence (70-90%) had been reached.

*Measurement of ROS levels.* Specific measurements of mitochondrial ROS levels were performed with some modifications as described before[98]. Briefly, cells were cultured on glass bottom dishes (27 mm; Nunc by Thermo Fisher Scientific, 150682) in DMEM with GlutaMAX™-I (low glucose, Gibco by life technologies, 21885) supplemented with 10% FBS and 1% penicillin-streptomycin (5000 U/ml). Either lutein (10, 20 or 40 μM; Sigma Aldrich PHR1699) or vehicle (0,1%, 0,2% or 0,4% dimethyl sulfoxide, DMSO, Sigma Aldrich D8418) was added to the culture medium and cells were cultured for 24, 48 and 72 h in a humidified atmosphere (95% air, 5% CO$_2$) at 37 °C. Next, cells were incubated with MitoSOX™ Red (5 μM, 10 min, 37 °C; Invitrogen/ThermoFisher, M36008). The red fluorescence was documented using an Axio Observer Z1 microscope (Zeiss) with the dehydroethidium filter set (F39-500, F48-515, F47-895; AHF Analysetechnik). A minimum of 100 cells were analyzed per condition. Images were analyzed and fluorescence intensity was quantified using ImageJ software (http://rsbweb.nih.gov/ij/).

### Statistical analysis

Data are represented either as mean ± SEM with corresponding data points or as box and whiskers plot as indicated in each figure legends. At least three independent biological replicas carried out in a blinded manner where possible. Statistical analyses were performed using either two-sided Student's *t*-test, one-way ANOVA for multiple comparisons, or two-way ANOVA followed by Tukey's multiple-comparisons test for multiple comparisons and time-points. GraphPad Prism 8 software was used for all statistical analysis to calculate significance. Exact p values are specified in every figure legend. For very small *p* values ($p < 0.0001$) GraphPad does not provide exact numbers, which are therefore reported as $p < 0.0001$ versus respective control conditions. Survival experiments were analyzed with Log-rank test using the Online Application for Survival analysis OASIS 2 (see supplementary methods).

### Reporting summary

Further information on research design is available in the Nature Research Reporting Summary linked to this article.

### Data availability

The GEO accession numbers for microarray data reported in this paper are "GSE144573" and "GSE144574". Source data are provided with this paper. "WormBase", Omim and "Pubmed" are publicly available databases. All other relevant data supporting the key findings of this study are available within the article and its Supplementary Information files or from the corresponding author upon reasonable request.

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

## Acknowledgements

We would like to thank Professor Tom Johnson at the University of Colorado Boulder and Professor Shane Rea (now at University of Washington) for making possible theinitial customized HMAD RNAi screening. We also would like to thank Alison Kell, Jenny Cho, and Alessandro Torgovnick for technical help with RNAi screen and lifespan; Professor Proksch at the Heinrich Heine University of Duesseldorf for the compound library used in the suppressor screen; Núria Benseny-Cases at MIRAS beamline in ALBA Synchrotron Light Source, Cerdanyola del Vallès in Barcelona, and Genis Rabost for technical assistance with FTIR measurements and analysis; Anthony Luz and Ian Ryde at Duke University for assistance with Seahorse XFe24 analysis and DNA damage assays respectively; Dirk Schwitters for qPCR on brains from WT and NDUFS4$^{-/-}$ mice. Finally, we thank the Caenorhabditis Genetics Center (funded by the National Institutes of Health Office of Research Infrastructure Programs: P40OD010440) as well as the National Bioresource Project (NBRP) for *C. elegans* strains and Professors Sieburth, Kaplan and Calahorro for providing additional mutants used in this work. The Wood-Whelan fellowship covered Silvia Maglioni costs to visit Joel Meyer Laboratory. This work was possible thanks to financial support from the German Research Foundation (DFG grants VE663-3/1 and VE663/8-1), the Federal Ministry of Education and Research (JPI-HDHL, Grant no. 01EA1602), and the Heinrich Heine University of Duesseldorf (Strategic Research Funding 2014) to NV; the National Institutes of Health (P42ES010356) to J.N.M., a fellowship from the China Scholarship Council (CSC201607030005) to Z.L.; the Spanish Ministry of Science, Innovation and Universities (RTI2018-096273-B-I00) and the 'Severo Ochoa' Programme for Centers of Excellence in R&D (SEV-2015-0496) to A.L.; the ERC Stg 337327 Mito-PexLyso to N.R.

## Author contributions

N.V. conceived and supervised the study. S.M. and N.V. designed the experiments. S.M., M.M., L.Z. and V.B. performed the experiments. S.M., N.V., M.M., L.Z., A.S., N.R. analyzed the data. S.M. and N.V. wrote the paper. F.D., A.L., N.R., J.M. edited the manuscript.

## Funding

## Competing interests

The authors declare no competing interests.
