## [Peer Review File · Nature Communications]

REVIEWER COMMENTS

Reviewer #1 (Remarks to the Author):

The authors use RNAi of nuo-5 and lpd-5 to describe a complex I depletion model of mitochondrial disease. The authors characterize the complex I deficiency model using biochemical and microarray analysis. Suppression of the neurodevelopmental deficits were screened for using a selection of compounds and the authors identify lutein as a possible therapeutic.

The authors state that they generated new *C. elegans* models for mitochondriopathies. The advancement from previous RNAi findings using the Ahringer library describing slow growth, sterile phenotypes listed on Wormbase is unclear.

The data presented lacks a clear story. For example, RNAi experiments are directed at targets ranging from mitochondrial dynamics to TCA cycle enzymes. However, the focus is on lpd-5 and nuo-5. The rationale behind the other mitochondrial targets or the focus on Leigh Syndrome vs another disease (e.g. optic dystrophy or Parkinson's Disease) is unclear. Overall, it is difficult to assess if the focus of the manuscript is on the development of new models of mitochondriopathies or if the focus is on lutein reversing a complex I deficiency.

The authors perform a suppressor screen on nuo-5 and lpd-5 worms. The details of the screen are missing. Was it an unbiased screen? What library are the chemical from? How were the compounds or doses chosen?

Some figures lack statistical analysis making the conclusions difficult to draw. Moreover, some statistical tests are using a one-way anova to analyze data sets with two genotypes and different treatment groups. Additionally, many of the experiments were from only two independent trials (N=2). The statistical methods could be improved to give the reader more confidence in the findings.

Is the rescue of lutein selective for nuo-5-mediated complex I deficiency or is this treatment general for all complex I deficiencies and/or all mitochondriopathies?

Minor:

Throughout the text, the authors refer to data as "very reproducible" or "considerable rescue", or "for the first time." The authors should refrain from this language and allow the reader to make conclusions based on the data and statistics.

The background on *C. elegans* as a model system in the introduction is extensive

Pg5 – line 141-149 – the authors suggest that systemic mitochondrial suppression has broader repercussion than neuronal silencing. However, the outputs are only looking at whole body effects, not necessarily a neuron specific outputs.

Pg 6 – line166 – Are the authors suggesting that the complex I N-module oxidises NADPH?

Some figures show the analysis of other strains (e.g. nuo-2, F53f4.5) and the significance of this data to the story is unclear.

Reviewer #2 (Remarks to the Author):

In this interesting study, the authors use *C. elegans* to develop models for mitochondrial diseases and most importantly use two of these models (Complex I mutants) for suppressor screens to identify potential substances that can be used as therapeutic tools.

I found the findings relevant for different reasons:

- 1) The authors analyze several mutants and show that while neurodevelopmental defects correlate with mitochondrial function, it is not the case with the longevity phenotype of these mutants, thus uncoupling the underlying signaling pathways.
- 2) They identify lutein as a novel compound with the ability (at least in worms) to rescue the worm developmental phenotype, and convincingly show that this rescue is independent from potential anti-oxidant effects.
- 3) Searching for the mechanism of lutein rescue, they identify an acetylcholine presynaptic alteration and a detrimental up-regulation of neuroligin, rescued by lutein supplementation, thus revealing a novel downstream effect of CI dysfunction at the synaptic level. Neuroligins and neuroreoxins upregulation was also confirmed in the brain of a *Ndufs4* mouse knock-out, supporting the conservation of this finding also in mammals.
- 4) Finally, they show that upregulation of neuroligin in the context of a mitochondriopathy is detrimental.

In conclusion, the work exploits the potentiality of *C. elegans* genetics to reveal a novel dysfunctional pathway in neurons deficient for Complex I and identify lutein as a potential therapeutic nutritional compound, thus having translational potential. This paper will stimulate further investigation in this respect in mammalian models.

Minor comments:

- 1) It is surprising that no major gene expression differences were observed under mild nuo-5 RNAi, but yet these animals are long-lived. What about previously identified pathways? The authors could comment in the discussion.
- 2) In Fig. S1B, please indicate what the is the difference among strains TU3401 and CL6114 to N2.
- 3) Fig. 4D, E. The plots show the percentage of associated genes to each CC component. However, is this statistical significant? More details would be important here.

4) The results of Fig. S10 not always match the text description. In Fig. S10A one can see an effect on lutein only in the *unc10*. The reduction in reporter expression does not seem so prominent in Fig.S10B-D from the pictures shown.

4) Why does lutein have so very different effects in mutants and in wild-type worms? This result is rather puzzling. The authors could comment on this result in the discussion.

Reviewer #3 (Remarks to the Author):

Maglioni et al. present their findings that lutein suppresses the developmental consequences of disrupting the Complex I subunits *nuo-5* and *lpd-5*. The group uses an impressively broad range of biochemical, genetic, molecular, cellular, and behavioral assays to investigate the neurodevelopmental effects of mitochondrial dysfunction. Interestingly, they find that lutein rescues the neurodevelopmental effects by restoring wild type levels of neuroigin expression. The findings are significant and the experiments are thorough. They raise the intriguing question of how lutein regulates neuroigin expression, but this question is beyond the scope of this manuscript. There are technical and statistical issues that need to be addressed to support the findings.

Comments:

- 1) The authors demonstrate that a reduction in pharyngeal pumping efficiency is disrupted in *nuo-5* and *lpd-5* animals, however the levels of lutein remain elevated compared to controls. It is not clear how the authors determined that lutein is taken up orally rather than absorbed through the cuticle.
- 2) It is difficult to compare the effectiveness of the different RNAi backgrounds in Fig S1B. The statistical significance of the difference in RNAi efficiency is not indicated. It is also not clear whether and how the authors took into account that the control grows more slowly in CL6114, which complicates comparing the developmental speed between it and the two other strains.
- 3) Line 242 – only lutein was effective against both RNAi. The results should be revised carefully to avoid implying that merely increasing the number of fertile adults to 20% represents a significant rescue by any of the compounds.
- 4) The effectiveness of lutein appears variable: significant rescue was seen with 1uM and 50uM concentrations but not with 10uM. It is more concerning that the effect of lutein on the development of *lpd-5* animals was not reproducible in Figure S7. Is there a difference between this assay and the one presented in Fig 3 that would explain the difference at day 6 between the two assays?
- 5) Whether lutein affected *lpd-5* RNAi should be tested in addition to *nuo-5*.
- 6) Almost twice as many genes were differentially regulated between wild type and strong *nuo-5* depleted animals in the second assay compared to 1152 genes between control and strong *nuo-5* suppression in the first assay. Presumably the only difference is the use of DMSO in the second assay. Why DMSO would have such a strong effect or how the two assays differ should be clarified and discussed.

Minor clarifications:

- 1) The nuo-5 and lpd-5 genes should be defined in the introduction.
- 2) How were the 30 natural compounds selected to be screened?
- 3) What is the R to E ratio?

** See Nature Research's author and referees' website at www.nature.com/authors for information about policies, services and author benefits.

POINT-BY-POINT REPLY LETTER TO REVIEWERS' COMMENTS

Reviewer #1 (Remarks to the Author):

The authors use RNAi of *nuo-5* and *lpd-5* to describe a complex I depletion model of mitochondrial disease. The authors characterize the complex I deficiency model using biochemical and microarray analysis. Suppression of the neurodevelopmental deficits were screened for using a selection of compounds and the authors identify lutein as a possible therapeutic.

The authors state that they generated new *C. elegans* models for mitochondriopathies. The advancement from previous RNAi findings using the Ahringer library describing slow growth, sterile phenotypes listed on Wormbase is unclear.

Thanks for giving us the possibility to clarify one of the innovative aspects of our work. The first aim of our study was indeed to generate new models for mitochondriopathies. To this end we customized a small sub-library of RNAi clones specifically targeting diseases-related genes, whose potency was titrated in different ways to better mimic disease progression and severity. The advancement compared to previous findings from large-scale RNAi using the Ahringer library are thus at least threefold:

- 1) Manual screen of a small number of clones and at different RNAi potency, allowed a wider characterization of animals' phenotypes: those previously described in wide-genome RNAi screen after 4 days (growth, fertility) plus additional ones, such as animals' movement, paleness, sickness, F1 progeny defects. Thus, our targeted approach led to the identification of different clones which resulted in the *Mit* phenotypes including some previously neglected in wide RNAi screening.
- 2) The selection of a small number of clones to look at in two consecutive generation and at different RNAi concentrations, allow us to identify a certain number of clones whose phenotypic features consistently correlate with the potency of the RNAi and thus better reflect disease severity. These clones were followed up on to analyze additional HMAD-relevant features such as lifespan, mitochondria/metabolic and neuronal alterations.
- 3) Moreover, our broader phenotypic characterization led us to reveal something which was overseen by previous analysis with these genes, as also nicely pointed out by the second reviewer "...while neurodevelopmental defects correlate with mitochondrial function, it is not the case with the longevity phenotype of these mutants, thus uncoupling the underlying signaling pathways".

These three points were touched already in the results session. The discussion has been now revised to reinforce above-mentioned innovative aspects which were not clearly spelled out in the original version of the manuscript: "*While we utilized phenotypic features already described by us and others with some of the targeted genes (e.g. nuo-2), such as slow growth and reduced fertility, our targeted approach allow us to screen for additional phenotypes, such as animals' paleness, sickness or defects in the progeny, and to deeply characterize selected clones in more details in terms of survival and neuro-metabolic features.*" (lines 494-498).

The data presented lacks a clear story. For example, RNAi experiments are directed at targets ranging from mitochondrial dynamics to TCA cycle enzymes. However, the focus is on *lpd-5* and *nuo-5*. The rationale behind the other mitochondrial targets or the focus on Leigh Syndrome vs another disease (e.g. optic dystrophy or Parkinson's Disease) is unclear.

Thanks for this comment which helped to further explain the rationale behind our genes' selections.

1) Initial 41 genes:

New text has been now added to clarify how we choose the initial 41 genes for our RNAi screen: "*To develop new models for mitochondriopathies we initially carried out a cross-reference research through public-available data (i.e. WormBase, OMIM, PubMed) to identify as many as possible C.*

elegans annotated genes orthologous to nuclear-encoded genes which when mutated in humans lead to severe mitochondrial dysfunction and consequent neurodevelopmental pathologies.” (lines 109-111).

2) Complex I related genes:

At the completion of the RNAi screen, we further phenotypically characterized different complex I related genes based on the “.....very strong and reproducible phenotype (L2/L3 larval arrest) and health importance in humans (disease prevalence)” as was already specified in the original version of the manuscript (lines 137-140).

3) nuo-5 and lpd-5:

The rationale for specifically focusing on *lpd-5* and *nuo-5* has now been clarified by restructuring the text of the second chapter: “For further characterization of the new models, we then narrowed our study to two CI subunits, which are both part of the CI functional N-module, the structure responsible for oxidation of NADH to NAD⁺ (the main CI activity), and specifically NUO-5 (NDUFS1 homolog) and LPD-5 (NDUFS4 homolog). Suppression of the two subunits gave similar phenotypes, but while.....” (lines 159-162).

Overall, it is difficult to assess if the focus of the manuscript is on the development of new models of mitochondriopathies or if the focus is on lutein reversing a complex I deficiency.

As summarized in the abstract and clearly appreciated by the reviewer, this is a very comprehensive study, which in fact focus on two important aspects. In the first part of the manuscript, we developed new models for mitochondriopathies and extensively characterized their biochemical and neurobehavioral features. In the second part, we carried out a compound screen in search of possible disease suppressors and identified a molecular target mechanistically involved in disease pathogenesis and in lutein protective effect. Additional studies in the lab are currently following up on both aspects of this work.

The authors preform a suppressor screen on *nuo-5* and *lpd-5* worms. The details of the screen are missing. Was it an unbiased screen? What library are the chemical from? How were the compounds or doses chosen?

The experimental details of the screen were explained in material and methods and in the workflow in **Supplementary Fig. 5**. Further details have been now included as follow. We primarily choose “...natural compounds (available in our laboratory and with previously described anti -aging, -bacterial, -inflammatory, -oxidant, -cancer activity; **Supplementary Table 5**) in search of potential disease suppressors.” (lines 233-234). Moreover, we selected a concentration range “...based on previous studies showing beneficial effects in different systems” (lines 248-249).

Some figures lack statistical analysis making the conclusions difficult to draw. Moreover, some statistical tests are using a one-way anova to analyze data sets with two genotypes and different treatment groups. Additionally, many of the experiments were from only two independent trials (N=2). The statistical methods could be improved to give the reader more confidence in the findings.

We really appreciated the reviewer has carefully checked our statistical analysis, which we have now systematically revised and improved according to the suggestions, as delineated in details in the next 3 points.

1) Figures lacking statistical analysis:

- Error bars were missing in **Fig. 8e** and have been now included;
- Some panels show data coming from microarray analysis (**Fig. 4d,e; 5d,e,f; 6g,h; 7b,c**) thus displaying GO terms or genes already selected for their significantly different changes. We now specified this important detail, were missing, in the corresponding figure legends. Microarray data are available online as pointed out in the “data availability” statement.

- We prefer not to include the statistics in the chemotaxis curve (**Fig. 3f,h**) since the panel would become too messy. Moreover, although we could display the data as bar graph for all the different time points and conditions (and include statistics for each point), we choose for simplicity to only show max chemotaxis (**Fig. 3g,i**).

2) Tests using one-way anova:

- The reviewer is right, for different panels (**Fig. 3d,e,g,i,k,l,m; Fig 6a,b,c,d,e,f; Fig 7d,f,g; Fig. 8a,b,c,d,e,g**) we have now recalculated the statistic using two-way anova. For most panels this did not change the significance. In panel **3m** the new test did not give significant difference in *Ndufs4* cells, but this was anyway barely significant even before so it did not change our interpretation of the results. In panels **6a** and **8b,g** the aldicarb sensitivity for one time point was not significant with the new analysis and we choose to show the complete time-course curves.
- Instead, we now specified in the figure legend for panel **3j** that the two disease models were assessed separately, therefore unpaired one-way anova was correctly carried out to be able to show the data in one graph.

3) Experiments with N=2 independent experiments:

Some of the experiments were indeed only repeated twice. However, due to the complexity in carrying out some of the assays with tiny arrested larvae or to the relevance of the specific readout, coupled to high number of animals used (and therefore small p values obtained), we reason it is enough to have N=2. Specifically:

- In the Mitosox experiments (**Fig. 2h; Fig. 3k**) the total number of animals is high and the test is very reproducible, leading to a very small standard error and very significant p-value <0,0001.
- *dct-1::GFP* quantification (**Fig. 2k**) also gave 2 very similar replicas leading to very small standard error. Plus, this is not a critical experiment for the overall message of the study.
- Pharyngeal pumping data with *lpd-5* RNAi (**Fig. 2c; Fig. 3j**) is a very tricky experiment and the standard error is low. Moreover, we primarily focus on *nuo-5* in the rest of the study, which in fact was assessed in triplicate.
- The aldicarb assay with the different anti-oxidants (**Fig. 7f**) was carried out in duplicate only for some of the conditions, others were in triplicate. We now included, as we did for every experiment, the number of animals for each condition. The error is small here as well.

Is the rescue of lutein selective for *nuo-5*-mediated complex I deficiency or is this treatment general for all complex I deficiencies and/or all mitochondriopathies?

In the original version of the manuscript, we had already shown that lutein suppresses different defects of both *nuo-5* and *lpd-5*-depleted animals (i.e. development, chemotaxis, ROS production, **Fig. 3e-l**). We now further tested the effect of silencing *lpd-5*, as well as two additional disease-related genes, *T20H4.5* and *F53F4.10*, on animals' resistance to aldicarb along with the rescuing effect of lutein. The new results are described in the text as follow: "*Remarkably, we observed the same Hic phenotype upon silencing of other disease-associated CI subunits, namely lpd-5, T20H4.5 and F53F4.1 (Table 1), and lutein significantly rescued the synaptic defect in all three disease models (Supplementary Fig. 10a-c).*" (lines 359-361).

Minor:

Throughout the text, the authors refer to data as "very reproducible" or "considerable rescue", or "for the first time." The authors should refrain from this language and allow the reader to make conclusions based on the data and statistics.

We appreciated this comment and carefully revised the text to avoid overinterpretation of our findings. Most of our statements, as delineated below, are based on interpretation of the new data compared to

the literature in the field and to our previous studies and we thus believe we are not overstating our findings.

1) “*very reproducible*” was used 3 times (lines 90, 125, 519), always referring to the phenotypes obtained upon suppression of different nuclear encoded mitochondrial proteins. We are working with mitochondria in *C. elegans* for the past 15 years and can confidentially state that we achieved very reproducible phenotypes.

2) “*considerably rescue*” was used only once (line 420) referring to the synaptic defect rescued by lutein. The rescuing effect was proved using different genetic and behavioral assays, which indeed support a significant rescue.

3) “*for the first time*” was used 4 times (lines 458, 468, 552, 625), always referring to the newly disclosed *nlg-1*-mediated synaptic dysfunction disclosed in the disease models, which, to our knowledge, has never been described before. We removed “*for the first time*” on line 606, where it was redundant.

The background on *C. elegans* as a model system in the introduction is extensive.

We have now slightly trimmed the *C. elegans* introduction. Nonetheless, since our study primarily uses *C. elegans*, we wanted to make sure to carefully underline major pro and cons of the model system to allow full understanding of the novelty and originality of the findings also for a not too specialized audience.

Pg5 – line 141-149 – the authors suggest that systemic mitochondrial suppression has broader repercussion than neuronal silencing. However, the outputs are only looking at whole body effects, not necessarily a neuron specific outputs.

Indeed, we wanted to specifically address here the broader non-neuronal (more systemic) impact of mitochondrial dysfunction on disease state. The use of the two strains with different neuronal sensitivity to silencing allowed us to reveal that systemic mitochondrial deficiency has a stronger effect on development than neuronal specific suppression. This also indicate that suppressing mitochondrial function at systemic level is a more appropriate way to model the disease than only suppressing it in the neurons. We clarify the text accordingly: “*Symptoms associated with CI-deficiency can be concurrently ascribed to cell-autonomous and non-autonomous effects of the mitochondrial dysfunction. Animals L3 arrest implies the RNAi is likely working in the nervous system – as in this stage mitochondria are necessary in the nervous system to progress through development – but C. elegans gene silencing can be less efficient in neurons. Thus, to address the contribution of neuronal vs non-neuronal cells to animals’ pathology,.....*” (lines 168-173).

Pg 6 – line166 – Are the authors suggesting that the complex I N-module oxidises NADPH?

The N-module binds and oxidizes NADH resulting in the liberation of electrons that are transferred via flavin mononucleotide (FMN) onto a series of Fe–S clusters. We mistakenly wrote NAD(P)H, now corrected into NADH (lines 160-161).

Some figures show the analysis of other strains (e.g. *nuo-2*, F53f4.5) and the significance of this data to the story is unclear.

As we explained in the introduction: “*Here, we thus exploited C. elegans to characterize early biochemical, cellular and neurobehavioral features resulting from severe complex I deficiency*” (lines 88-89). In the first part of the Result section we describe an initial screening with 41 *C. elegans* genes all related to mitochondriopathies. We silenced these genes through RNAi in a first phenotypic screen and afterwards we focused only on genes coding for mitochondrial CI subunits to narrow down our research (lines 137-140). The genes mentioned by the reviewer (e.g. *nuo-2*, F53f4.5) are listed and underlined in Table I, and, as specified in the text, they all belong to CI. In an early stage of our study, we characterized sensory and locomotory defects in animals upon RNAi against of different CI subunits

(Supplementary Fig. 2a-e). We then only focused on *nuo-5* and *lpd-5* among those to narrow down the study: “For further characterization of the new models, we then narrowed our study to two CI subunits, which are both part of the CI functional N-module, the structure responsible for oxidation of NADH to NAD⁺ (the main CI activity), and specifically NUO-5 (NDUFS1 homolog) and LPD-5 (NDUFS4 homolog)” (lines 159-162).

Reviewer #2 (Remarks to the Author):

In this interesting study, the authors use *C. elegans* to develop models for mitochondrial diseases and most importantly use two of these models (Complex I mutants) for suppressor screens to identify potential substances that can be used as therapeutic tools.

I found the findings relevant for different reasons:

- 1) The authors analyze several mutants and show that while neurodevelopmental defects correlate with mitochondrial function, it is not the case with the longevity phenotype of these mutants, thus uncoupling the underlying signaling pathways.
- 2) They identify lutein as a novel compound with the ability (at least in worms) to rescue the worm developmental phenotype, and convincingly show that this rescue is independent from potential anti-oxidant effects.
- 3) Searching for the mechanism of lutein rescue, they identify an acetylcholine presynaptic alteration and a detrimental up-regulation of neuroligin, rescued by lutein supplementation, thus revealing a novel downstream effect of CI dysfunction at the synaptic level. Neuroligins and neuroreceptors upregulation was also confirmed in the brain of a *Ndufs4* mouse knock-out, supporting the conservation of this finding also in mammals.
- 4) Finally, they show that upregulation of neuroligin in the context of a mitochondriopathy is detrimental.

In conclusion, the work exploits the potentiality of *C. elegans* genetics to reveal a novel dysfunctional pathway in neurons deficient for Complex I and identify lutein as a potential therapeutic nutritional compound, thus having translational potential. This paper will stimulate further investigation in this respect in mammalian models.

Minor comments:

- 1) It is surprising that no major gene expression differences were observed under mild *nuo-5* RNAi, but yet these animals are long-lived. What about previously identified pathways? The authors could comment in the discussion.

Thanks for this interesting comment. We were actually also surprised. As pointed out in the discussion (lines 509-513), we hypothesized that major transcriptomic changes were not observed in the L3 upon mild mitochondrial suppression since changes associate with longevity take longer to occur. Indeed, transcriptomic data in search of aging signatures in *C. elegans*, are normally collected in young or gravid adult, including those in previously described long-lived *Mit* mutants. Therefore, while severe mitochondrial suppression already leads to important transcriptomic genes in early L3 larvae (which will then arrest their development), gene reprogramming necessary to promote health-span may become evident later during development to compensate the mild mitochondrial stress. We hope this help explaining our results.

- 2) In Fig. S1B, please indicate what the is the difference among strains TU3401 and CL6114 to N2.

We now added the genotype of the strains in the figure legend of **Supplementary Fig. 2g** “... specifically: *TU3401: sid-1(pk3321); uls69[myo2p::mCherry + unc-119p::sid-1]*, a neuronal specific sensitive strain, and *CL6114: nre-1(hd20) lin-15b(hd126)*, a strain with overall increased sensitivity to RNAi.” and referred to them in the text (line 175-176), making clear the differences with the wild-type (N2) strain.

3) Fig. 4D, E. The plots show the percentage of associated genes to each CC component. However, is this statistical significant? More details would be important here.

Thank you for pointing that out. The statistical test used was reported in material and method as well in the figure legend. Nonetheless we missed to specify that, in these panels, we were showing only statistically significant GO terms (with corrected p-value < 0,05). We now added this important detail in the corresponding figure legend: “...only statistically significant GO terms are shown, corrected p-values < 0.05.” (line 1191-1192). We also added a supplementary excel table (**Supplementary Data 1**) with statistic of Gene Ontology (GO) analysis of the microarray experiment shown in **Fig. 4**.

4) The results of Fig. S10 not always match the text description. In Fig. S10A one can see an effect on lutein only in the *unc10*. The reduction in reporter expression does not seem so prominent in Fig.S10B-D from the pictures shown.

Please note that Fig S10 is now named **Supplementary Fig. 11**. We improved the figure presentation, which, along with the following explanations, will hopefully help reconciling the seemingly observed discrepancies. **Supplementary Fig. 11a** shows the qPCR results while **Supplementary Fig. 11b-d** show representative pictures of GFP reporter strains. It is important to note that the reporter strains used here express GFP in individual neurons, therefore the differences may not be striking at a first look (and may not correspond to changes revealed by the qPCR analysis) but are highly significant when quantified with image J. From the qPCR only *unc-10* was statistically significant but the trend is visible also for the other genes. In **Supplementary Fig 11b** we now added a dashed circle around the posterior pharyngeal bulb, where the induction of expression upon *nuo-5*, and its reduction after lutein treatment, is more appreciable. In **Supplementary Fig 11c** *glb-10* is visibly reduced upon *nuo-5* silencing and goes back to control level with lutein treatment. In **Supplementary Fig. 10d** *unc-17* was quantified only in the second proximal head neuron (as explained in supplementary methods), which is now highlighted with an arrow. We also corrected the name of 2 panels (**Supplementary Fig. 11b, c**) to which we were referring in the text in the paragraph “Quantification of Gene Expression through fluorescent transgene reporters” in supplementary methods.

4) Why does lutein have so very different effects in mutants and in wild-type worms? This result is rather puzzling. The authors could comment on this result in the discussion.

Thanks a lot for this very important observation. This is indeed true and actually further support the specificity of Lutein in the disease model. Most importantly, this advocate for the safety of using Lutein in CI-deficiency possibly without “side effects”. To further emphasize this very relevant concept we have now added a couple of sentences at the very beginning of the discussion (lines 473-477): “In this regard, it is also very important to note that, at least at the dose used in this study, lutein had no effects in the wild-type animals. This indicates lutein specifically targets pathogenetic mechanisms induced by CI-deficiency, with no “side effects” in a non-compromised background, a clear positive indication for the development of new potential therapeutic strategies”.

Reviewer #3 (Remarks to the Author):

Maglioni et al. present their findings that lutein suppresses the developmental consequences of disrupting the Complex I subunits *nuo-5* and *lpd-5*. The group uses an impressively broad range of biochemical, genetic, molecular, cellular, and behavioral assays to investigate the

neurodevelopmental effects of mitochondrial dysfunction. Interestingly, they find that lutein rescues the neurodevelopmental effects by restoring wild type levels of neuroligin expression. The findings are significant and the experiments are thorough. They raise the intriguing question of how lutein regulates neuroligin expression, but this question is beyond the scope of this manuscript. There are technical and statistical issues that need to be addressed to support the findings.

Comments:

1) The authors demonstrate that a reduction in pharyngeal pumping efficiency is disrupted in *nuo-5* and *lpd-5* animals, however the levels of lutein remain elevated compared to controls. It is not clear how the authors determined that lutein is taken up orally rather than absorbed through the cuticle.

Thanks for this very interesting comment. The reviewer is absolutely right in that we did not determine how exactly lutein is taken up by the animals. As the reviewer points out, the pharyngeal pumping defect implies that routes other than oral uptake, e.g. through the skin or via sensory neurons exposed to the environment, may concur to its absorption and consequent systemic effect. We added a comment on this in the corresponding result session (lines 265 and 271).

2) It is difficult to compare the effectiveness of the different RNAi backgrounds in Fig S1B. The statistical significance of the difference in RNAi efficiency is not indicated. It is also not clear whether and how the authors took into account that the control grows more slowly in CL6114, which complicates comparing the developmental speed between it and the two other strains.

As the reviewer correctly indicated, CL6114 grows slightly slower than N2. Nonetheless we took that into account and therefore normalized the effect of the RNAi among each strain with its internal empty vector control treated animals.

3) Line 242 – only lutein was effective against both RNAi. The results should be revised carefully to avoid implying that merely increasing the number of fertile adults to 20% represents a significant rescue by any of the compounds.

We agree with the reviewer and specified (line 256) that 20% of animals developing into adults is a cut-off we arbitrarily choose, which allowed us to use an automatically quantifiable phenotype to select compounds for follow up studies.

4) The effectiveness of lutein appears variable: significant rescue was seen with 1uM and 50uM concentrations but not with 10uM. It is more concerning that the effect of lutein on the development of *lpd-5* animals was not reproducible in Figure S7. Is there a difference between this assay and the one presented in Fig 3 that would explain the difference at day 6 between the two assays?

We noticed indeed difference of compounds efficacy from one experiment to another, which could in time ascribe to different variables such as room temperature, type of plates, change in chemicals media and compound supply (batch), freshness of the compounds, efficiency of solubility and therefore bioavailability. Unfortunately, despite we try to reproduce experimental conditions in each trial, there are always subtle difference we may not always control which can affect the results. This is why we decided to use very selective criteria to choose our “best” suppressor treatment. **Fig. 3b** and **Supplementary Fig. 7a** are coming from two different experiments, the former from the screening in 6-well plates, while the latter is a day-by day development assay on single plates. As mentioned above, small changes in the experimental setting led to variability in the compounds’ efficacy. We now repeated 2 times the experiment in **Supplementary Fig. 7a** and were able to show a similar result to the one shown in **Fig. 3b**, thus avoiding confusion for the readers. The fact that we could achieve similar

results using different technical experimental settings in the end strongly supports the biological significance of our data.

5) Whether lutein affected *lpd-5* RNAi should be tested in addition to *nuo-5*.

In this manuscript we primarily focused our attention on *nuo-5* and clearly showed that lutein does not affect *nuo-5* RNAi efficacy (microarray, **Fig. 5c**, as well as qPCR, **Fig. 3d**). Moreover, we showed that lutein does not reduce the effect of unrelated RNAi (such as *skn-1*, **Supplementary Fig. 7b**). Thus, we believe it is very unlikely that Lutein may negatively affect the potency of *lpd-5* or of other RNAi. Nonetheless, in our ongoing follow up studies we are focusing on further testing Lutein and other potential suppressors on *lpd-5* and other CI subunits-depleted animals. We are therefore optimizing qPCR conditions to check whether the compounds affect the efficacy of the specific RNAi (which is not trivial as the primers in most cases recognize the RNA coming from the bacterial fed for the silencing experiments). Moreover, to further support the specific effect of the compounds (independently from their effects on gene silencing) we are generating knock-in disease models. These are ongoing studies which we would prefer to keep separate from this manuscript, which is instead primarily focusing on lutein and *nuo-5*.

6) Almost twice as many genes were differentially regulated between wild type and strong *nuo-5* depleted animals in the second assay compared to 1152 genes between control and strong *nuo-5* suppression in the first assay. Presumably the only difference is the use of DMSO in the second assay. Why DMSO would have such a strong effect or how the two assays differ should be clarified and discussed.

In all conditions DMSO was used as vehicle, in fact, the sample showed in **Fig. 4** (con and strong) partially overlap with samples shown in **Fig. 5**. We now changed the color-code of the Venn Diagram in **Fig. 5a** to avoid misunderstanding. The number of genes exclusively altered in the comparison con vs *nuo-5* strong are 1152 in one comparison (Fig. 4a) and 1044 in the other (Fig. 5a), therefore very similar. In the Venn Diagram of **Fig. 4a** the over 2000 genes are shared with the mild vs strong comparison, condition which is absent in **Fig. 5**.

Minor clarifications:

1) The *nuo-5* and *lpd-5* genes should be defined in the introduction.

Thanks for the suggestion. We have now briefly introduced them: “*NUO-5* (*NDUFS1* homolog) and *LPD-5* (*NDUFS4* homolog) are both part of the CI functional N-module, the structure responsible for the complex oxidoreductase activity. Numerous mutations in these two submits, leading to reduced protein expression and/or CI activity, have been described in human [20].” (lines 94-97). We also implemented their description in the second chapter of the work, before we start their further characterization: “For further characterization of the new models, we then narrowed our study to two CI subunits, which are both part of the CI functional N-module, the structure responsible for oxidation of NADH to NAD⁺ (the main CI activity), and specifically *NUO-5* (*NDUFS1* homolog) and *LPD-5* (*NDUFS4* homolog). While their suppression gave similar phenotypes...” (lines 159-162).

2) How were the 30 natural compounds selected to be screened?

As also mentioned above to reply to one of the comments from Reviewer #1, we have now specified that we primarily choose natural compounds “...available in our laboratory, which were previously shown to have anti -aging, -inflammatory, -oxidant and -cancer activities” (lines 233-234) and we selected a concentration range “...based on previous studies showing beneficial effects in different systems” (line 248-249).

3) What is the R to E ratio?

As reported in the legend of **Supplementary Fig. 13b** “*The R/E ratio is the ratio of the repolarization and depolarization waves in each pump (R and E waves) and is a parameter that indicates probable alterations in the normal muscle contraction/relaxation.*” We now added a sentence in the text to point out that: “**Supplementary Fig. 13b** shows a representative EPG with details of each parameter” (line 673).

REVIEWER COMMENTS

Reviewer #1 (Remarks to the Author):

The authors have addressed my comments.

Reviewer #2 (Remarks to the Author):

The authors have answered the questions raised and addressed them in the revised manuscript.

Reviewer #3 (Remarks to the Author):

Concerns remain regarding 3 of the original comments:

Comment 2. I do not see any revisions that correspond to the response to this comment. The data remains the same as in the original figure and the normalized data does not seem to be included. A statistical comparison of the normalized data is still needed to support the statement on line 179-181.

Comment 3. While choosing an arbitrary cutoff of 20% is a reasonable approach to choosing candidates for follow up, the text incorrectly implies that anything above 20% is a significant difference. Specifically, the statements on line 256 "...11 partially prevented the developmental arrest induced by nuo-5 RNAi" and on line 257 "...7 rescued that of lpd-5 RNAi" misrepresent the data. The words prevented and rescued can only be used if the differences are statistically significant compared to the nuo-5 RNAi on DMSO and lpd-5 RNAi + DMSO conditions. As stated later in the paragraph, these data show that only 2 treatments prevented/rescued the nuo-5 developmental arrest phenotype and 3 treatments prevented/rescued the lpd-5 developmental arrest phenotype. Perhaps a comparison of earlier stages for the other compounds would reveal more significant changes? Otherwise, the statements should be removed. Error bars for lutein 10 uM appear to be missing.

Comment 4. Line 264-265 "observed that the effect of lutein was the most reproducible and significant" should be changed to indicate that it was the only reproducible and significant treatment. The inability to replicate the phenotypes of isovitexin and macrosporin indicate they do not suppress the developmental defects.

** See Nature Research's author and referees' website at www.nature.com/authors for information about policies, services and author benefits.

Reviewer #3 (Remarks to the Author):

We would like to thank this Reviewer for giving us the possibility to further improve our manuscript by addressing the 3 left-open comments. As described below, we now modified the manuscript as requested. New text in the main manuscript (line 253 and lines 260-261) as well as in the Supplementary file (Supplementary Fig. 1g) have been highlighted in yellow for simplicity.

Concerns remain regarding 3 of the original comments:

Comment 2. I do not see any revisions that correspond to the response to this comment. The data remains the same as in the original figure and the normalized data does not seem to be included. A statistical comparison of the normalized data is still needed to support the statement on line 179-181.

To avoid confusion related to the intrinsic strain developmental timing, we have now modified the graph showing only two categories of development (fully or not fully developed) and as suggested we included statistics. The % changes in animals' development induced by the RNAi were normalized to the stage reached by each strain treated with empty vector control (con), which have been now settled to 100% "fully developed". The data clearly indicate that the *nuo-5* and *lpd-5* RNAi have a stronger impact on development of the CL6114.

Comment 3. While choosing an arbitrary cutoff of 20% is a reasonable approach to choosing candidates for follow up, the text incorrectly implies that anything above 20% is a significant difference. Specifically, the statements on line 256 "...11 partially prevented the developmental arrest induced by *nuo-5* RNAi" and on line 257 "...7 rescued that of *lpd-5* RNAi" misrepresent the data. The words prevented and rescued can only be used if the differences are statistically significant compared to the *nuo-5* RNAi on DMSO and *lpd-5* RNAi + DMSO conditions.

As stated later in the paragraph, these data show that only 2 treatments prevented/rescued the *nuo-5* developmental arrest phenotype and 3 treatments prevented/rescued the *lpd-5* developmental arrest phenotype. Perhaps a comparison of earlier stages for the other compounds would reveal more significant changes? Otherwise, the statements should be removed. Error bars for lutein 10 uM appear to be missing.

Thanks for the comment. We have now included the error bar for lutein 10uM. We agree with the reviewer that comparisons at earlier stage could reveal more significant changes. However, discriminating between small (L1-L2) larvae development is much more subjective to error and might introduce additional caveats when it comes to screening many compounds. We stated that "Among tested conditions, 11 partially prevented the developmental arrest induced by *nuo-5* RNAi, while 7 rescued that of *lpd-5* RNAi". To further tune down our statement we included partially (line 253) also for those affecting *lpd-5* RNAi development. The word partially, by no means implies significance. Indeed, we also state in the next sentence that "Significant differences were achieved only with isovitexin 10µM on *nuo-5*, with kahalalide F 0,5µM on *lpd-5* and with lutein 1µM on both" to specifically clarify that while a total of 12 conditions allowed more than 20% of worms to develop into fertile adults after 6

days of treatment (our arbitrary cutoff), only 3 conditions induced significant changes.

Comment 4. Line 264-265 “observed that the effect of lutein was the most reproducible and significant” should be changed to indicate that it was the only reproducible and significant treatment. The inability to replicate the phenotypes of isovitexin and macrosporin indicate they do not suppress the developmental defects.

As indicated, we changed the sentence to better describe the findings “...observed that despite a general trend, the treatment with lutein was the only one reproducibly showing significant effects” (lines 260-261). The rescuing effect of isovitexin and macrosporin is actually visible but does not reach statistical difference (this does not mean that their biological effect is absent).

REVIEWER COMMENTS

Reviewer #3 (Remarks to the Author):

Specifically, regarding comment 2 and Supplementary Figure 2, the authors state: *we assessed the effect of nuo-5 and lpd-5 silencing on development in a neuronal specific sensitive strain (TU3401) and in a strain with overall increased sensitivity to RNAi (but especially in neurons) (CL6114). Interestingly, the effect of the RNAi was reduced in the first strain while it was increased in the second one (Supplementary Fig. 2g),*

Two issues remain. The first is that I do not believe the authors can conclude that RNAi has a weaker or stronger effect in TU3401 or CL6114 animals compared to control animals unless the data are directly compared to the data from the control animals (using statistics). To make those conclusions, the effect of nuo-5 and lpd-5 RNAi on CL6114 and TU3401 animals must be compared to the effect of nuo-5 and lpd-5 RNAi on control worms.

The second is that comparing developmental speed between the strains is complicated because the strains all grow at different speeds to begin with. The current changes to the graph are misleading, since the CL6114 animals grow more slowly than N2 or TU3401 animals even in the absence of RNAi. The term fully developed cannot be used to simultaneously describe the not yet gravid animals in CL6114 and the gravid animals in N2 and TU3401. I suggest the authors either find an accurate way to normalize the data (perhaps using time to YA as a constant measurement across all strains) and then compare between strains or remove the figure and associated comments from the manuscript.

For comment 3, I suggest the authors specify the treatments meet their arbitrary cutoff of 20% and avoid making any comparison to wild type controls unless the difference is significant. Therefore, 'partially prevented' and 'partially rescued' should be removed.

For comment 4, since the isovitexin and macrosporin experiments were repeated and still did not reach significance, the authors should remove the clause "observed that despite a general trend..." and specify that only the lutein treatment was reproducible. If the authors want to report a biological effect of isovitexin and macrosporin, they need to use a quantifiable assay that demonstrates their difference from controls.

** See Nature Research's author and referees' website at www.nature.com/authors for information about policies, services and author benefits.

Point-by-point reply letter

We appreciate reviewer' comments, which we have now addressed as requested (underlined in light blue in the revised manuscript files).

Reviewer #3 (Remarks to the Author):

Specifically, regarding comment 2 and Supplementary Figure 2, the authors state: *we assessed the effect of nuo-5 and lpd-5 silencing on development in a neuronal specific sensitive strain (TU3401) and in a strain with overall increased sensitivity to RNAi (but especially in neurons) (CL6114). Interestingly, the effect of the RNAi was reduced in the first strain while it was increased in the second one (Supplementary Fig. 2g),*

Two issues remain. The first is that I do not believe the authors can conclude that RNAi has a weaker or stronger effect in TU3401 or CL6114 animals compared to control animals unless the data are directly compared to the data from the control animals (using statistics). To make those conclusions, the effect of nuo-5 and lpd-5 RNAi on CL6114 and TU3401 animals must be compared to the effect of nuo-5 and lpd-5 RNAi on control worms.

The second is that comparing developmental speed between the strains is complicated because the strains all grow at different speeds to begin with. The current changes to the graph are misleading, since the CL6114 animals grow more slowly than N2 or TU3401 animals even in the absence of RNAi. The term fully developed cannot be used to simultaneously describe the not yet gravid animals in CL6114 and the gravid animals in N2 and TU3401. I suggest the authors either find an accurate way to normalize the data (perhaps using time to YA as a constant measurement across all strains) and then compare between strains or remove the figure and associated comments from the manuscript.

We agree with the reviewer that comparing the effect of the RNAi on these strains is not easy, but we had clearly explained the differences to avoid conveying misleading conclusions. Nonetheless, the reviewer is right that as represented before, the differences of the RNAi effects in the CL6114 and TU3401 animals, compared to the effect in the WT strain, were not statistically significant. Thus, to better match the observed biological effects we have now represented the data in a different way and only displayed numbers of animals reaching the fertile stage (once all reached that stage in the respective control conditions). We cannot use time to YA (as suggested by the reviewer) since animals do not reach this stage in every condition (as they arrest at early stages). We have accordingly also implemented the description in the text (lines 174-185) to avoid misinterpretation of the results.

For comment 3, I suggest the authors specify the treatments meet their arbitrary cutoff of 20% and avoid making any comparison to wild type controls unless the difference is significant. Therefore, 'partially prevented' and 'partially rescued' should be removed.

We have modified the sentence as requested (lines 257-258).

For comment 4, since the isovitexin and macrosporin experiments were repeated and still did not reach significance, the authors should remove the clause “observed that despite a general trend...” and specify that only the lutein treatment was reproducible. If the authors want to report a biological effect of isovitexin and macrosporin, they need to use a quantifiable assay that demonstrates their difference from controls.

We have modified the sentence as requested (lines 264-265).

REVIEWERS' COMMENTS

Reviewer #3 (Remarks to the Author):

The authors have addressed my concerns.

** See Nature Portfolio's author and referees' website at www.nature.com/authors for information about policies, services and author benefits